# Calibrating calving parameterizations using graph neural network emulators: application to Helheim Glacier, East Greenland

**Younghyun Koo**[1,2,3], **Gong Cheng**[4], **Mathieu Morlighem**[4], and **Maryam Rahnemoonfar**[1,2]

[1]Department of Computer Science and Engineering, Lehigh University, Bethlehem, PA 18015, USA
[2]Department of Civil and Environmental Engineering, Lehigh University, Bethlehem, PA 18015, USA
[3]National Snow and Ice Data Center (NSIDC), Cooperative Institute for Research in Environmental Sciences (CIRES), University of Colorado Boulder, Boulder, CO 80309, USA
[4]Department of Earth Sciences, Dartmouth College, Hanover, NH 03755, USA

**Correspondence:** Maryam Rahnemoonfar (maryam@lehigh.edu)

**Abstract.** Calving is responsible for the retreat, acceleration, and thinning of numerous tidewater glaciers in Greenland. An accurate representation of this process in ice sheet numerical models is critical to better predict the future response of the ice sheet to climate change. While traditional numerical models have been used to simulate ice dynamics and calving under specific parameterized conditions, the computational demand of these models makes it difficult to efficiently fine-tune these parameterizations, adding to the overall uncertainty in future sea level rise. In this study, we adopt three standard graph neural network (GNN) architectures, including graph convolutional network, graph attention network, and equivariant graph convolutional network (EGCN), to develop surrogate models for finite-element simulations from the Ice-sheet and Sea-level System Model. GNNs are particularly well-suited for this problem as they naturally capture the representation of unstructured meshes used by finite-element models. When these GNNs are trained with numerical simulations of Helheim Glacier, Greenland, for different calving stress thresholds, they successfully reproduce the observed evolution of ice velocity, ice thickness, and ice front migration between 2007 and 2020. Moreover, these emulators exhibit uncertainties of less than 10 %–20 % when extrapolating to out-of-sample calving parameterization cases. Among the three GNN architectures, EGCN outperforms the others by preserving the equivariance of graph structures. By leveraging the GPU-based GNN emulators, which are 30–34 times faster than traditional numerical simulations, we determine the temporal variations of the optimal calving threshold that minimizes the misfit between modeled and observed ice fronts. This fine-tuned calving parameterization, enabled by GNN emulators, can enhance the reliability of numerical models in capturing glacier mass loss driven by calving.

## 1 Introduction

Over the past three decades, the Greenland ice sheet has experienced an average annual loss of 170 Gt of ice, resulting in a global mean sea level rise exceeding 15 mm (Otosaka et al., 2023). This trend of mass loss has intensified in recent years. Between 1990 and 2000, the annual mass loss hovered around $40\,\mathrm{Gt\,a^{-1}}$, but in the 2010s, it surged to approximately $280\,\mathrm{Gt\,a^{-1}}$ (Otosaka et al., 2023; Mouginot et al., 2019). This escalating mass loss can be attributed to two primary processes: (1) the change in surface mass balance driven by enhanced surface melt and (2) calving and submarine melting of marine-terminating glaciers, commonly referred to as frontal ablation (King et al., 2020; Choi et al., 2021). In specific regions and seasons, ice discharge may be responsible for more than 50 % of the total ice sheet mass loss (Mouginot et al., 2019; King et al., 2020; Choi et al., 2021; Aschwanden et al., 2019).

Enhanced calving significantly impacts ice dynamics, often resulting in ice flow acceleration and subsequent thinning (Bondzio et al., 2017; Cheng et al., 2022; Lippert et al., 2024). Given that calving is sensitive to climate conditions (Greene et al., 2024; Wood et al., 2021), it is important to

understand how future calving rates would impact ice sheet mass balance and sea-level rise (Choi et al., 2021). Numerous studies have utilized ice sheet numerical models to explore the applicability of various calving laws to Greenland and Antarctic ice sheets and identified the set of model parameters that minimize the misfit between model and observations (Wilner et al., 2023; Choi et al., 2018). According to Choi et al. (2018), the von Mises calving law (Morlighem et al., 2016, VM;) best replicates observed terminus positions of nine outlet glaciers in Greenland compared to other existing laws.

Although numerical models can provide reliable solutions for ice flow when ice extent is kept constant, capturing the precise impacts of spatiotemporally varying calving rates on terminus migration and ice flow in numerical models remains challenging. Additionally, assimilating remote sensing observations into numerical models, required to infer certain model parameters, is both complex to implement and computationally intensive (Choi et al., 2023). Furthermore, the integration of calving as a boundary condition in numerical models introduces significant complexity since calving directly alters the ice geometry and model domain during simulations. Consequently, identifying the optimal calving parameterizations consistent with observations is difficult and time-consuming, thereby limiting our ability to project the future mass balance of ice sheets under various parameter settings (Choi et al., 2018; Edwards et al., 2021; Morlighem et al., 2020).

To address the computational demands of numerical models, various statistical approaches have emerged relying on faster machine learning models in lieu of ice sheet numerical models. While traditional numerical models often necessitate using high-performance computing clusters to solve partial differential equations (PDEs) on central processing units (CPUs), machine learning emulators offer the advantage of operating on lighter computational resources, leveraging the parallel processing capabilities of graphic processing units (GPUs). For instance, Downs et al. (2023) used a Gaussian process (GP) emulator to infer the sensitivity of time-independent model parameters to the frontal ablation of Helheim Glacier in Southeast Greenland. Although they were able to identify the best set of calving threshold parameters in the VM calving law, their GP approach did not account for spatial relationships or interactions between neighboring nodes. Additionally, their emulator focused on matching the observed and modeled terminus positions along a central flow line rather than the entire glacier system.

Given that ice dynamics and calving are affected by the glacier geometry and underlying bed topography, it is important for the emulators to learn the spatial context across the entire glacier domain. To account for spatial relationships between nodes for emulating ice sheet dynamics, graph neural networks (GNNs) have emerged as an effective neural network architecture for handling irregular non-Euclidean data structures such as molecular structures, point clouds,

social networks, and natural language (Zhang et al., 2019). GNNs are adaptable to any type of data structure organized as graphs, comprising *nodes* (i.e., data points) and *edges* (i.e., the connections between nodes), which make them suitable to manifest meshes of numerical simulations (Pfaff et al., 2021). GNNs make predictions by utilizing pairwise message passing between nodes, wherein information exchange occurs, updating individual node features through interactions with connected nodes. Inspired by the resemblance of the mesh structure in finite-element analysis to a graph structure, numerous studies have investigated the training of emulators of finite-element numerical simulations using GNNs (Fu et al., 2023; Shivaditya et al., 2022; Black and Najafi, 2022; Perera et al., 2022; Salehi and Giannacopoulos, 2022; Maurizi et al., 2022; Jiang and Chen, 2023).

In this study, we investigate whether GNNs can be used as the backbone architecture for statistical mapping between input and output variables of finite-element ice sheet modeling while embedding the spatial connections between nodes. GNN emulators take direct advantage of unstructured meshes of the Ice-sheet and Sea-level System Model (ISSM), allowing flexible spatial resolution and efficient allocation of computational resources, and can speed up model parameter search. Here, as an illustration, we use GNN emulators to determine how calving parameterization affects ice dynamics while maintaining the accuracy and precision of ISSM simulations. Our focus is on Helheim Glacier, chosen as the target site for training and evaluating our GNN emulators (Fig. 1a). Helheim Glacier is one of the largest outlet glaciers in Greenland, and it has been shown that its ice velocity is closely correlated to the position of its terminus (Cheng et al., 2022). Despite considerable advancements in observation and modeling techniques over the past decades, the mechanisms governing Helheim Glacier's ice front position remain elusive (Bevan et al., 2015; Miles et al., 2016; Cheng et al., 2022). A key contributing factor to this knowledge gap is the computational intensity of numerical models incorporating calving, impeding the fine-tuning of calving parameters to accurately reflect observations. As the main parameter that determines the terminus positions, the temporal variations of calving parameters and their impacts on calving should be explored (Downs et al., 2023). In this study, we train GNN models using simulation data derived from numerical models and evaluate their fidelity and computational efficiency in modeling the dynamics and calving front migration of Helheim Glacier. We assess the potential of GNN architectures as statistical emulators for numerical finite-element ice sheet models to represent spatial features of ice sheet dynamics and calving across the entire glacier domain.

The remainder of the paper is structured as follows. Section 2 provides a comprehensive review of relevant literature about calving parameterization in ice sheet models, other machine-learning emulators employed in ice sheet modeling, and the use of GNNs as emulators for finite-element models. In Sect. 3, we describe the training data collected from

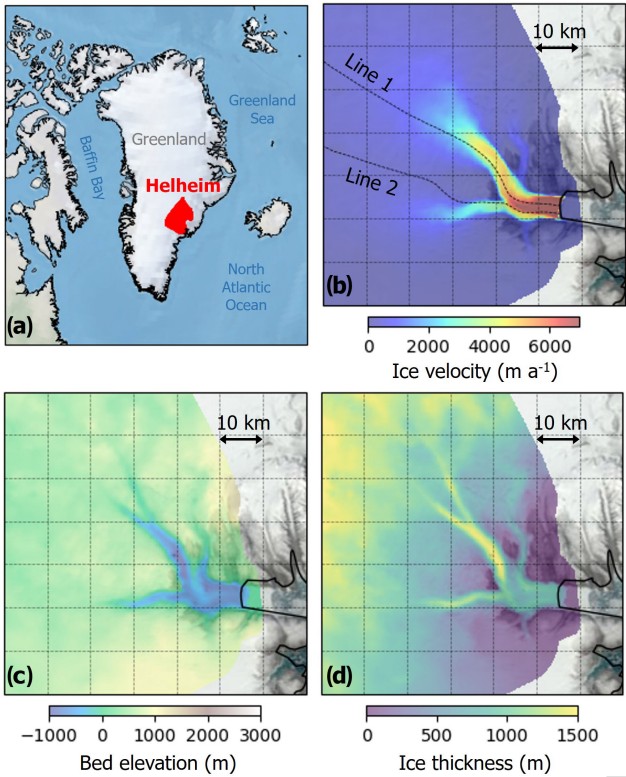

**Figure 1. (a)** Location of Helheim Glacier, Greenland; **(b)** ice velocity; **(c)** bed elevation; and **(d)** ice thickness near the calving front of Helheim Glacier in 2007. The background image is the MODIS true color image from 20 June 2023.

numerical simulations, while Sect. 4 outlines the specifics of GNN architectures. Section 5 presents the accuracy and computational efficiency of these models in replicating the dynamics of Helheim Glacier. In addition, we demonstrate the utility of GNN emulators in optimizing calving parameterization to match simulation results with observations. Finally, we discuss the scientific implications of calving calibration and the limitations of GNN emulators in Sect. 6.

## 2 Background

### 2.1 von Mises calving law

In the VM calving law, the calving rate, $c$, is assumed to be proportional to both the tensile stress and the magnitude of ice velocity, formulated as follows (Morlighem et al., 2016):

$$c = ||\boldsymbol{v}|| \frac{\tilde{\sigma}}{\sigma_{\max}}, \tag{1}$$

where $\tilde{\sigma}$ is a scalar quantity representing the effective tensile stress of the ice, $\sigma_{\max}$ is a stress threshold that needs to be calibrated on a glacier-by-glacier basis, and $\boldsymbol{v}$ is the ice flow velocity at the ice front. The migration rate of the ice front is

then determined from the calving rate $c$ using the following equation (Morlighem et al., 2016):

$$\boldsymbol{v}_{\mathrm{f}} = \boldsymbol{v} - (c + \dot{M})\boldsymbol{n}, \tag{2}$$

where $\boldsymbol{v}_{\mathrm{f}}$ is the ice front migration rate, $\dot{M}$ is the melting rate on the calving front, and $\boldsymbol{n}$ is a unit normal vector pointing outward from the ice domain. In the ISSM, the numerical ice sheet model we use in this study, moving boundaries of ice sheets are represented implicitly using the level set method (Osher and Sethian, 1988; Bondzio et al., 2016; Morlighem et al., 2016; Cheng et al., 2024). The level set method defines a scalar field $\varphi(\boldsymbol{x}, t)$: $\varphi$ is negative in the location of $\boldsymbol{x}$ that is ice-covered at time $t$, positive where there is no ice, and the zero contour of $\varphi$ defines the ice boundary. With a given initial condition $\varphi(\boldsymbol{x}, t_0)$, the level set $\varphi$ is updated by solving an advection equation using the migration rate $\boldsymbol{v}_{\mathrm{f}}$ that

$$\frac{\partial \varphi}{\partial t} + \boldsymbol{v}_{\mathrm{f}} \cdot \nabla \varphi = 0. \tag{3}$$

In VM, $\sigma_{\max}$ is the only parameter that needs to be calibrated (Morlighem et al., 2016). A lower value of $\sigma_{\max}$ correlates to weaker ice and, consequently, a larger calving rate. According to Morlighem et al. (2016), $\sigma_{\max}$ is consistent with the range of ice tensile strength, which is typically around 1 MPa but may be as low as 0.7 MPa and not exceed 3 MPa. In most of the previous studies, this $\sigma_{\max}$ value is simply assumed to remain constant over time within a predefined range. However, environmental conditions around the ice front (e.g., temperature, mélange, sea ice, ocean wave, tide) can vary over time, potentially influencing this calving parameter in response and leading to substantial changes in calving patterns (de Juan et al., 2010; Todd and Christoffersen, 2014; Kondo and Sugiyama, 2023; Xie et al., 2019; Wehrlé et al., 2023). Therefore, fine calibration of $\sigma_{\max}$ is essential for accurately reproducing observed glacier evolution.

### 2.2 Machine learning emulator for ice sheet modeling

Machine learning techniques have been extensively employed as statistical emulators for numerical ice sheet models and further for calibration of model parameters. Tarasov et al. (2012) applied Markov chain Monte Carlo methods to artificial neural networks (ANNs) for the Bayesian calibration of model parameters. This approach allows for obtaining posterior distributions for different model parameters given observational datasets. Similarly, Brinkerhoff et al. (2021) used an ANN surrogate model to infer the posterior distributions for parameters that govern the behavior of the sliding law and the hydrological model. Downs et al. (2023) employed the GP for the calving parameterization associated with the VM calving law in Helheim Glacier. Nevertheless, the ANNs of these previous studies are simple feedforward networks without spatial embeddings between nodes. Given that ice dynamics are determined by the topographical features, it is

essential for the surrogate models to fully capture the interactions between neighboring elements to accurately represent realistic ice sheet behavior.

In replicating spatial characteristics of glaciers, convolutional neural networks (CNNs) have been predominantly adopted as the main architectures based on their advantages in capturing the spatial variations in topographical features of glaciers (Jouvet et al., 2022; Jouvet and Cordonnier, 2023; Jouvet, 2023; Verjans and Robel, 2024). For instance, the CNN of the Instructed Glacier Model (IGM) (Jouvet et al., 2022) reproduced the ice dynamics from the Parallel Ice Sheet Model (Winkelmann et al., 2011, PISM) and CfsFlow models (Jouvet et al., 2008). Jouvet (2023) extended this CNN emulator to address inversion problems, inferring optimal ice thickness distribution, ice flow velocity, and ice surface elevation to match both a Stokes model and observational data. Another CNN emulator introduced by Jouvet and Cordonnier (2023) employed a physics-informed loss function to minimize the energy associated with ice-flow equations during training. However, CNNs cannot take full advantage of finite-element ice sheet models on their native grid because they rely on regular grids (Zhang et al., 2019). Given the GNN's capacity to account for dynamic interactions between nodes and edges (Satorras et al., 2022), it can be a promising tool for predicting the dynamic behavior of ice. Indeed, Koo and Rahnemoonfar (2025) demonstrated the fidelity and computational efficiency of a GNN architecture in emulating a finite-element ice sheet numerical model. This capability of GNN in preserving the mesh structure from the numerical models can have significant advantages in accurately describing ice front migration.

## 2.3 Graph neural networks for finite-element analysis

GNNs have been broadly used to emulate finite-element numerical models due to the similarities between computational meshes and graphs. Perera et al. (2022) developed a GNN framework to simulate fracture and stress evolution in brittle materials, training their model with data generated from a finite-element method fracture solver. Similarly, Shivaditya et al. (2022) proposed a GNN surrogate model for finite-element simulations of metal forging processes, demonstrating superior performance compared to other machine learning models and achieving a 10-fold reduction in processing time. Salehi and Giannacopoulos (2022) also developed PhysGNN, a GNN framework tailored for simulating soft tissue deformation, and Maurizi et al. (2022) utilized GNNs to predict stress, strain, and deformation across various material systems, including fiber and stratified composites, and lattice metamaterials. Fu et al. (2023) proposed a boundary-oriented graph embedding (BOGE) approach within the GNN framework for solving finite-element cantilever beam problems, incorporating both boundary elements and local neighbor elements. Jiang and Chen (2023) introduced a novel graph attribute representation for triangular meshes in finite-element von Mises stress problems, effectively capturing geometry and boundary conditions to mitigate over-smoothing issues associated with deep GNNs. Black and Najafi (2022) introduced a multi-fidelity GNN for the cantilever beam problem, leveraging low-fidelity projections to inform high-fidelity modeling across arbitrary subdomains of subgraphs. However, despite the prevalence of finite-element analysis in ice sheet modeling (Larour et al., 2012; dos Santos et al., 2021; Gagliardini et al., 2013), to the best of our knowledge, GNNs have yet to be adopted exclusively for modeling calving dynamics and calibrating parameters involved.

## 3 Data

### 3.1 Ice sheet numerical simulation

To generate training datasets for the GNN emulators, we conduct transient simulations of ice dynamics and calving of Helheim Glacier between 2007 and 2020, using ISSM (Larour et al., 2012). The shelfy-stream approximation (SSA; MacAyeal, 1989) is used for describing ice flow. The SSA, which assumes depth-independent horizontal velocity and negligible vertical shear stresses, is appropriate for fast-flowing glaciers controlled by basal sliding such as the Helheim Glacier (Cheng et al., 2022; Choi et al., 2018).

The model setup is identical to the one described in Cheng et al. (2022). A two-dimensional unstructured mesh is constructed with a spatial resolution ranging from 100 m in the fast-flowing ice front to 1500 m in the inland domain, ultimately comprising 46 434 elements and 23 466 vertices (nodes). The transient simulations run forward in time with a time step of 1.825 d (0.005 years), and we output the state of the model every 10 time steps ($\sim$ 18 d). Consequently, each transient simulation generates a total of 261 outputs between 2007 and 2020. Basal friction is calibrated and held constant using the surface velocities from satellite interferometry (Mouginot et al., 2017, 2019) (Fig. 1b). Bed topography and the initial ice thickness are from BedMachine Greenland v6 (Morlighem et al., 2017) (Fig. 1c and d). Surface mass balance (SMB) is from the Regional Atmosphere Model (Tedesco and Fettweis, 2020), and the ocean thermal forcing is from Wood et al. (2021). The melting rate at the calving front (i.e., $\dot{M}$ in Eq. 2) is parameterized based on Rignot et al. (2016). To examine the sensitivity of ice dynamics to $\sigma_{\max}$ of the VM calving law, we run transient solutions for 9 different $\sigma_{\max}$ values (i.e., 0.70, 0.75, 0.80, 0.85, 0.90, 0.95, 1.00, 1.05, and 1.10 MPa) based on the values proposed by Choi et al. (2018).

The ISSM simulations provide the solutions of ice velocity, ice thickness, and a mask of ice-covered region every 10 time steps. We convert the triangular mesh from ISSM into a data structure that aligns with the input and output requirements of deep learning architectures. Specifically, for GNN architectures, we convert the meshes into graph nodes

and edges by extracting adjacency matrices that represent the connectivity between nodes. In a triangular mesh, each element consists of three nodes that are interconnected by edges (Fig. 2). Using the nodes and elements of the mesh en-5 sures that the resolution of this graph matches exactly with the finite-element mesh used in ISSM simulations.

## 3.2 Observations

To determine the best $\sigma_{\max}$ that aligns with real observations, we compare the numerical simulation results with remote-10 sensing derived terminus positions and ice velocities for the same periods as the numerical simulations (2007–2020). First, we use surface ice velocities with a spatial resolution of 150 m (Mouginot et al., 2017, 2019), which have a precision better than $20\,\mathrm{m\,a^{-1}}$ (Mouginot et al., 2017). Second, 15 we use a time series of calving front positions of Helheim Glacier from Greene et al. (2024). During the targeted time frame from 2007 to 2020, we use monthly averaged ice front positions, yielding a total of 156 distinct ice front positions for analysis.

## 20 4 Method

Ice sheet modeling can be regarded as a node-regression problem within graph structures, where the output features of individual nodes are derived from the input features of nodes. The unstructured meshes of ISSM can be represented 25 as graph structures, with node connectivity expressed via adjacency matrices. Based on the graph structures of the ISSM meshes, we develop three GNN architectures: graph convolutional network (GCN), graph attention network (GAT), and equivariant graph convolutional network (EGCN). Typical 30 GNN architectures update graph nodes iteratively through message-passing processes between neighboring nodes, and the way to achieve this message passing determines the specific type of GNN architecture. For the undirected graph $\mathcal{G} = (\mathcal{V}, \mathcal{E})$ with $N$ nodes $v_i \in \mathcal{V}$, edges $(v_i, v_j) \in \mathcal{E}$ TS1, and 35 an adjacency matrix $A \in \mathbb{R}^{N \times N}$, $l$th GNN layer receives a set of node features $\mathbf{h}^{(l)} = \{h_1^{(l)}, h_2^{(l)}, \ldots, h_N^{(l)}\}, h_i^{(l)} \in \mathbb{R}^{F_l}$, as the input and produces a new set of node features, $\mathbf{h}^{(l+1)} = \{h_1^{(l+1)}, h_2^{(l+1)}, \ldots, h_N^{(l+1)}\}, h_i^{(l+1)} \in \mathbb{R}^{F_{l+1}}$, for the next $l+1$th layer. $F_l$ and $F_{l+1}$ are the number of features in each node 40 at $l$th layer and $l+1$ layer, respectively. The GCN, GAT, and EGCN operate on graph structures but use different message-passing approaches in updating $\mathbf{h}^{(l+1)}$ from $\mathbf{h}^{(l)}$. By comparing three representative GNN architectures, we evaluate what approach is more effective in replicating ice sheet dynamics 45 and calving from the ISSM simulations.

## 4.1 Graph convolutional network

First, we employ a GCN proposed by Kipf and Welling (2017). We design a GCN with one input layer, five graph convolutional hidden layers, and one output layer (Fig. 2). The number of hidden layers is determined after conducting 50 trial and error experiments with several options: we tested 1, 2, 5, and 10 hidden layers, and 5 hidden layers showed the best accuracy. The graph convolutional hidden layers are inspired by the localized first-order approximation of spectral graph convolutions on graph-structured data (Kipf and 55 Welling, 2017). For each graph convolutional layer, the number of features is set to 128. Similar to the hidden layers, the number of hidden features was determined from trial and error with 4 options: 32, 64, 128, and 256. The weights of graph convolutional layers are updated via the layer-wise propaga-60 tion rule as follows:

$$h_i^{(l+1)} = \sigma \left( \sum_{j \in \mathcal{N}(i)} \frac{1}{c_{ij}} \mathbf{W}^{(l)} h_j^{(l)} \right), \quad (4)$$

where $\mathcal{N}(i)$ is the set of neighbors of node $i$, $c_{ij}$ is an appropriately chosen normalization constant for the edge $(v_i, v_j)$ defined as the product of the node degrees (i.e., $c_{ij} = 65$ $\sqrt{|\mathcal{N}(j)|}\sqrt{|\mathcal{N}(i)|}$), and $\mathbf{W}^{(l)} \in \mathbb{R}^{F_{l+1} \times F_l}$. $\mathbf{W}^{(l)}$ is a layer-specific trainable weight matrix ($\mathbf{W}^{(l)} \in \mathbb{R}^{F_{l+1} \times F_l}$), and $\sigma(\cdot)$ is an activation function; we use the Leaky ReLU activation function with a 0.01 negative slope in this study.

## 4.2 Graph attention network  70

Since the original GCN filters merely depend on graph structures and node connectivity (Eq. 4), a model trained with a certain graph structure can have limitations in general applicability to different graph structures. The GAT is proposed to address such shortcomings by adding masked self-attention 75 layers (Veličković et al., 2018). This architecture assigns different weights to different nodes in a neighborhood by inserting a self-attention mechanism in a hidden layer, which can allow better generalizability for different ice conditions or topographies. The propagation process in graph attention lay-80 ers can be expressed by the following equation (Veličković et al., 2018):

$$h_i^{(l+1)} = \sigma \left( \sum_{j \in \mathcal{N}(i)} \alpha_{ij}^{(l)} \mathbf{W}^{(l)} h_j^{(l)} \right), \quad (5)$$

where $\alpha^{(l)}$ is the attention score between node $i$ and node $j$ defined as follows: 85

$$\alpha_{ij}^{(l)} = \mathrm{softmax}_j \left( e_{ij}^{(l)} \right) = \frac{\exp\left( e_{ij}^{(l)} \right)}{\sum_{k \in \mathcal{N}(i)} \exp\left( e_{ik}^{(l)} \right)}$$

$$e_{ij}^{(l)} = a \left( \mathbf{W}^{(l)} h_i^{(l)}, \mathbf{W}^{(l)} h_j^{(l)} \right), \quad (6)$$

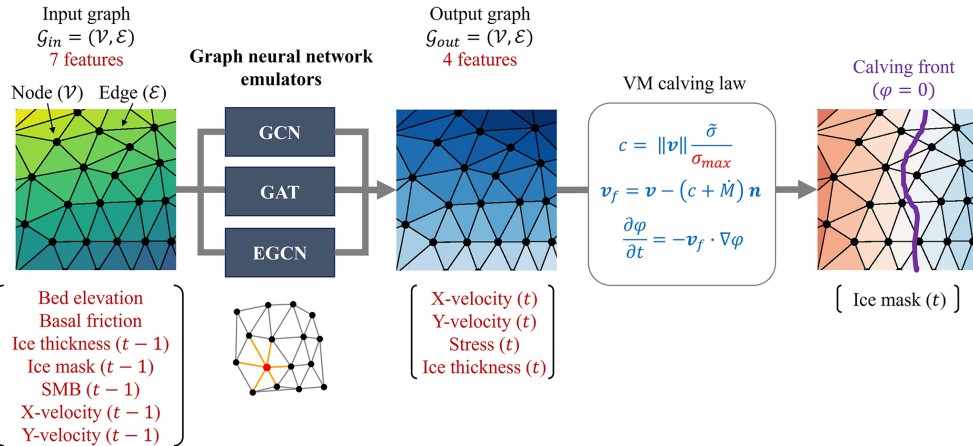

**Figure 2.** Schematic workflow of using GNN emulators to predict calving front.

where $a : \mathbb{R}^{F_{l+1}} \times \mathbb{R}^{F_{l+1}} \to \mathbb{R}$ is a self-attention mechanism to compute attention coefficient $e_{ij}^{(l)}$. This attention mechanism $a$ is a single-layer feedforward neural network parameterized by a weight vector $\boldsymbol{a} \in \mathbb{R}^{2F_{l+1}}$, normalized by LeakyReLU function afterward. The graph structure is applied to this attention mechanism by computing $e_{ij}^{(l)}$ for only nodes $j \in \mathcal{N}(i)$, where $\mathcal{N}(i)$ is set of neighbors of node $i$. We execute three independent attention mechanisms of Eq. (5) and average them for the final graph attention layer (Veličković et al., 2018). Similar to the GCN, the GAT consists of one input layer, five graph attention hidden layers with 128 features, and one output layer (Fig. 2).

### 4.3 Equivariant graph convolutional network

Another graph neural network we adopt is EGCN, which is designed to conserve equivariance to rotations, translations, reflections, and permutations in a graph structure (Satorras et al., 2022). Since our emulator is intended to predict ice front migration, we anticipate that the preservation of equivariance to rotations and translations on spatial coordinates via the EGCN structure guarantees sufficient generalizability to various graph structures of dynamics systems. An equivariant graph convolutional layer can be expressed by the following equations:

$$m_{ij} = \phi_e \left( h_i^{(l)}, h_j^{(l)}, \left\| x_i^{(l)} - x_j^{(l)} \right\|^2, a_{ij} \right) \tag{7}$$

$$v_i^{(l+1)} = \phi_v \left( h_i^{(l)} \right) v_i^0 + C \sum_{j \neq i} \left( x_i^{(l)} - x_j^{(l)} \right) \phi_x(m_{ij}) \tag{8}$$

$$x_i^{(l+1)} = x_i^{(l)} + v_i^{(l+1)} \tag{9}$$

$$m_i = x_i^{(l)} + C \sum_{j \neq i} m_{ij} \tag{10}$$

$$h_i^{(l+1)} = \phi_h \left( h_i^{(l)}, m_i \right), \tag{11}$$

where $a_{ij}$ is the edge attributes, $x_i$ and $x_j$ are the 2D coordinate embeddings for node $i$ and $j$, respectively, and $C$ is a constant for normalization computed as $1/|\mathcal{N}(i)|$. For the edge attributes, we use five attributes TS2 that can be extracted from the connecting nodes: distance, surface slope, base slope, and gradient of the $x$ and $y$ components of the ice velocity. $\phi_e$, $\phi_x$, and $\phi_h$ are the edge position and node operations, respectively, TS3 which are approximated by single-layer neural networks with 128 hidden features. Herein, the $x$ and $y$ components of ice velocities are represented as velocity embeddings ($v_i$), which induce the displacement of coordinate embeddings ($x_i$) (Eq. 9), and ice thickness and stress represented as hidden embeddings ($h_i^{(l)}$). More details about the EGCN architecture are described in Satorras et al. (2022).

### 4.4 Model training

From the ice sheet simulation results, we collect 2349 graphs (261 sets of results per transient simulations $\times$ 9 $\sigma_{\mathrm{max}}$ values). Each graph consists of 23 466 nodes and 139 798 edges. Individual nodes have numerical input and output features from the ISSM simulations and are connected to each other via edges, which are determined by the node connections of the finite elements. The input features include 7 variables that determine the ice sheet dynamic conditions: bed elevation, basal friction, ice thickness at time $t-1$, ice mask at time $t-1$, surface mass balance at time $t-1$, and $x$ and $y$ components of ice velocity at time $t-1$. The output includes 4 variables that represent ice dynamics at time $t$: the two components of ice velocity, ice thickness, and tensile stress $\tilde{\sigma}$ (Fig. 2). Our emulators are trained to establish a statistical mapping between input and output features: how the input variables at time $t-1$ change the output variables at time $t$ in numerical modeling. Previous machine learning studies attempted similar types of mapping via various deep learning architecture (Brinkerhoff et al., 2021; Downs et al., 2023; Tarasov et al., 2012; Jouvet and Cordonnier, 2023;

Jouvet et al., 2022; Jouvet, 2023). All input and output features are normalized to the range $[-1, 1]$ for stable learning using the nominal maximum and minimum values of these variables. After the GNNs predict ice velocity, ice thickness, and tensile stress $\tilde{\sigma}$, the ice mask at $t$ is computed from these outputs using a given $\sigma_{max}$ using Eqs. (1)–(3) (Fig. 2).

All 2349 graph structures are divided into training and test datasets based on the $\sigma_{max}$ values to assess whether our emulators can be generalized for different $\sigma_{max}$ values. We conduct four sets of sensitivity tests to evaluate whether the emulator can be reliably generalized to unseen $\sigma_{max}$ values (Table 1). In set A, the training dataset consists of samples with $\sigma_{max}$ values of 0.75, 0.90, and 1.05 MPa, while the remaining $\sigma_{max}$ values are used for testing. This set evaluates whether the model can represent the full range of $\sigma_{max}$ values when trained with evenly sampled $\sigma_{max}$ values. In set B, the training dataset includes low $\sigma_{max}$ values (0.70, 0.75, and 0.80 MPa) to test whether the model can generalize to high-value samples when trained only with low-value samples. In set C, we use medium $\sigma_{max}$ values (0.85, 0.90, and 0.95 MPa) as the training dataset to check whether the model can represent high- or low-value samples when trained only with medium-value samples. Finally, set D contains only high $\sigma_{max}$ values (1.00, 1.05, and 1.10 MPa) to check the model performance when extrapolating to lower values. For each sensitivity test, the training and test datasets contain 783 and 1566 graph structures, respectively. The model is optimized using the Adam stochastic gradient descent algorithm with the mean square error (MSE) loss function, over 500 epochs, and a learning rate of 0.001.

## 4.5 Model evaluation

We evaluate the ability of our emulators to reproduce ice velocity, ice thickness, and calving front migration by comparing them to ISSM simulation results. For this evaluation, we calculate two metrics: (i) root mean square error (RMSE) and (ii) binary calving front error (BE); RMSE is used to evaluate ice velocity and ice thickness predictions, and BE is used to evaluate the calving front delineation. These metrics are calculated using the following equations:

$$\mathrm{RMSE}(\hat{y}, y) = \sqrt{\frac{1}{N} \sum_{i=1}^{N} (\hat{y}_i - y_i)^2} \tag{12}$$

$$\mathrm{BE}(\hat{y}, y) = 1 - \frac{1}{N} \sum_{i=1}^{N} \mathbf{I}(\hat{y}_i = y_i), \tag{13}$$

where $\hat{y}$ denotes predicted values, $y$ denotes true values, $N$ is the number of data points, and $\mathbf{I}(\hat{y}_i = y_i)$ denotes the indication function that returns 1 if $\hat{y}_i = y_i$ and returns 0 if $\hat{y}_i \neq y_i$. RMSE represents the difference between the prediction and reference; a lower RMSE corresponds to better fidelity. In calculating BE, we convert the ice mask into binary values, representing ice nodes as 1 and non-ice nodes as 0. There-

fore, BE is a metric indicating the proportion of incorrectly predicted nodes relative to the total number of nodes; a lower BE corresponds to better fidelity.

## 5 Results

### 5.1 Model fidelity

Our GNN models are trained with four different training/test sets, as outlined in Table 1. The training and test errors of ice velocity, ice thickness, and calving front position for the GNN emulators compared to ISSM simulation results are shown in Fig. 3. When the GNN emulators are trained with even-value samples (set A), the training and test errors are almost the same for ice velocity, ice thickness, and calving front. Indeed, $t$ test results confirm no significant difference between training and test errors for all models (Table A1). This result indicates that the emulators trained with even-value samples can effectively represent the full range of $\sigma_{max}$ values.

In the case of low-value samples (set B), the test error is occasionally even lower than the training error, particularly in ice velocity prediction, which suggests that the emulators trained only with low-value samples can represent high $\sigma_{max}$ value cases with $< 10\%$ of uncertainty in most cases (Table A1). Although the test errors are slightly greater than training errors in sets C and D, their differences are primarily within a 5 % uncertainty range for ice velocity and ice thickness and within a 20 % uncertainty range for the calving front (Table A1). The results of sets C and D demonstrate that the GNN emulators trained with medium or high $\sigma_{max}$ values can generally have $< 10\%$–20 % of uncertainty in representing the out-of-sample $\sigma_{max}$ values. Overall, the sensitivity tests with four different sets show that GNN emulators trained with selected $\sigma_{max}$ values can be generalizable for all other $\sigma_{max}$ values with $< 10\%$–20 % of uncertainties even if those $\sigma_{max}$ values are not included in the training dataset. While the GP-based emulator by Downs et al. (2023) was limited to interpolating within the range of training samples, our GNN emulators can extrapolate to unseen $\sigma_{max}$ values. This capability in extrapolation is particularly valuable for optimizing calving parameterization beyond the scope of training $\sigma_{max}$ values.

Among the three GNN architectures, the EGCN exhibits notably lower RMSE values: $\sim 50$–60 m a$^{-1}$ of ice velocity RMSE and $\sim 30$ m of ice thickness RMSE (Fig. 3). That is, the EGCN reduces ice velocity RMSE by 40–150 m a$^{-1}$ and ice thickness RMSE by 15–20 m compared to the other GNN architectures (see Table A2 for statistical significance). On the other hand, the EGCN and GAT yield similar ice front predictions with BE ranging from 0.60 % to 0.75 %. The relatively lower errors achieved by the EGCN support that its equivariance architecture is beneficial in predicting ice dynamics. As mentioned in Sect. 4, the GCN, GAT, and EGCN

**Table 1.** Division of training/test $\sigma_{max}$ in four different sets.

| Set | Training $\sigma_{max}$ (MPa) | Test $\sigma_{max}$ (MPa) | Note |
|---|---|---|---|
| A | [0.75, 0.90, 1.05] | [0.70, 0.80, 0.85, 0.95, 1.00, 1.10] | Even-value samples |
| B | [0.70, 0.75, 0.80] | [0.85, 0.90, 0.95, 1.00, 1.05, 1.10] | Low-value samples |
| C | [0.85, 0.90, 0.95] | [0.70, 0.75, 0.80, 1.00, 1.05, 1.10] | Medium-value samples |
| D | [1.00, 1.05, 1.10] | [0.70, 0.75, 0.80, 0.85, 0.90, 0.95] | High-value samples |

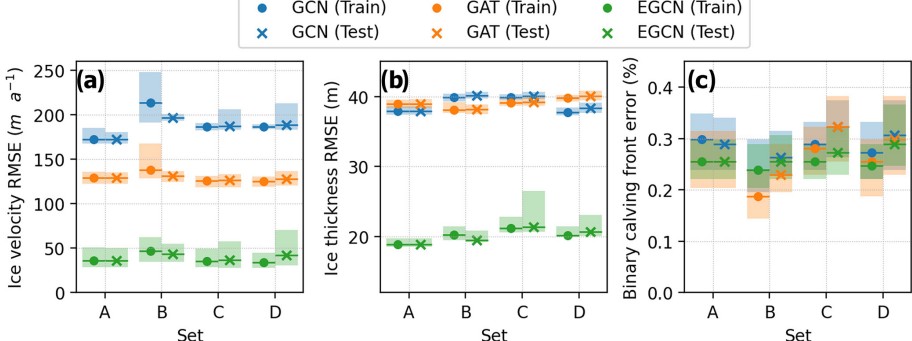

**Figure 3. (a)** RMSE of ice velocity, **(b)** RMSE of ice thickness, and **(c)** binary calving front error (BE) for four different sets. For each set, nine different $\sigma_{max}$ datasets are divided into training (three $\sigma_{max}$ values) and test (six $\sigma_{max}$ values) datasets as described in Table 1. The circle markers indicate the median errors from training datasets, and the crossing markers indicate the median errors from test datasets. The shaded areas indicate the interquartile ranges (i.e., 25 % quantile to 75 % quantile) of errors.

have distinct characteristics in message-passing approaches: the GCN simply uses the adjacency status between neighboring nodes to determine the weights during the propagation process (Eq. 4); the GAT uses additional self-attention mechanisms to evaluate the relative importance of neighboring nodes (Eq. 6); the EGCN uses the message passing from all nodes to preserve the equivariance of the entire graph (Eqs. 7–9). We conjecture that the equivariance architecture of the EGCN contributes significantly to the improvement of model fidelity.

Figures 4 and 5 show the maps of ice velocity and thickness on two $\sigma_{max}$ test cases of 0.70 and 1.10 MPa predicted by the GNN emulators trained with the training set A. Overall, all GNN emulators replicate the spatial and temporal patterns of ice velocity and thickness successfully. Additionally, they reproduce the trend of ice front retreat at lower $\sigma_{max}$ and advance at higher $\sigma_{max}$. In terms of ice velocity (Fig. 4), the EGCN shows the lowest errors of approximately $< 100 \, \mathrm{m \, a^{-1}}$ across the glacier domain, followed by GAT and GCN. In terms of ice thickness (Fig. 5), the EGCN generally shows an average RMSE $< 40$ m across the glacier domain, but the GCN and GAT have higher RMSEs of around $> 50$ m in some areas.

## 5.2 Computational time

The most significant advantage of GNN emulators is their ability to reduce this computation time by leveraging GPUs while maintaining finite element structures, which is not

completely available via other non-graph neural network architectures (Koo and Rahnemoonfar, 2025). When we run the ISSM numerical model, it takes 948 s ($\sim 15.8$ min) on average to complete one 13 year transient simulation for a given $\sigma_{max}$ value. This computation time of ISSM reflects the total elapsed time on a single node of the Texas Advanced Computing Center (TACC) Frontera supercomputing cluster, which is equipped with 56 cores of Intel 8280 Cascade Lake CPUs (192 GB memory). To evaluate the computational efficiency of the GNN emulators, we also record the time required to generate a single transient simulation for a given $\sigma_{max}$ value with our GNN emulators (Table 2). The GNN computations are performed on a CPU (Intel(R) Core(TM) i7-11700F; 32 GB memory) and a GPU (NVIDIA GeForce RTX 3070; 24 GB memory) within the same desktop (Lenovo Legion T5 26IOB6). We observe a dramatic speed-up with GPU-based deep learning emulators, achieving computation times 30–34 times faster than ISSM simulations. The GCN shows the highest speed-up, around 34 times faster than ISSM. Among the three GNN architectures, the GAT shows the longest computational time due to the complexity of its attention mechanisms. The EGCN also requires a longer computational time than the GCN because it operates on all graph nodes to preserve the equivariance in the entire graph, whereas the GCN operates only on the adjacent nodes. It is also noteworthy that using GPUs reduces the computation time of deep learning emulators by up to $\sim 4$ times compared to using CPUs.

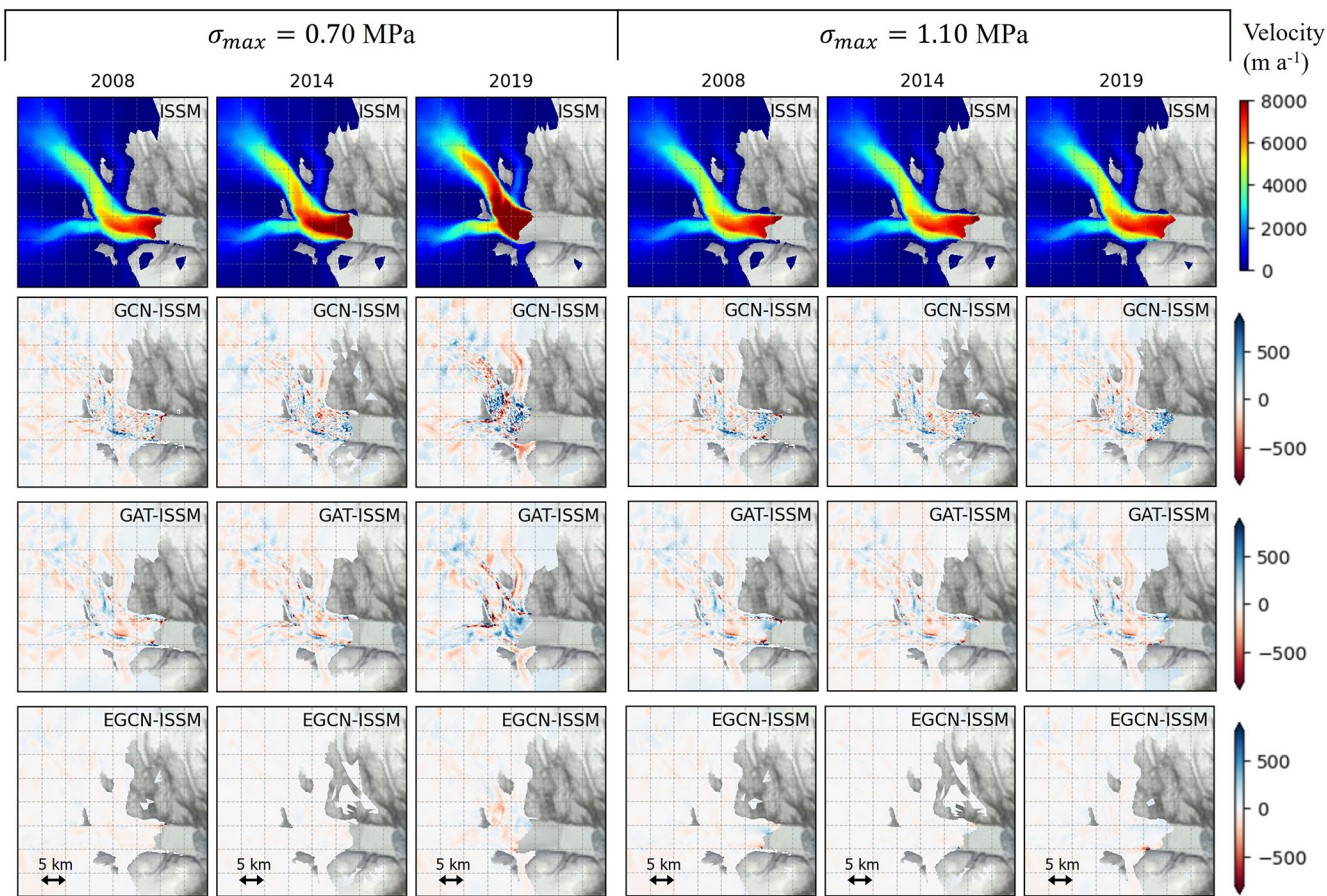

**Figure 4.** Maps of ISSM-simulated ice velocity and difference in ice velocity between GNN emulators and ISSM simulations (GCN, GAT, and EGCN from top to bottom) for 0.70 and 1.10 MPa test $\sigma_{\mathrm{max}}$ values of set A.

**Table 2.** Computation time for running one 13-year transient simulation from ISSM and GNN emulators and upstreaming computational time for training deep learning models. The shortest elapsed time and training time are highlighted in bold.

| Model | Averaged elapsed time for one transient run (speed-up compared to ISSM) | | Training time | Number of learnable parameters |
|---|---|---|---|---|
| | CPU | GPU | | |
| GCN | 45 s ($\times$21.0) | **28 s ($\times$33.8)** | **617 s (10.3 min)** | 67 588 |
| GAT | 135 s ($\times$7.0) | 31 s ($\times$30.6) | 1315 s (21.9 min) | 205 572 |
| EGCN | 65 s ($\times$14.6) | 29 s ($\times$32.7) | 814 s (13.6 min) | 104 066 |

This experiment is also promising because, while ISSM computation is performed on a supercomputer, GNNs can be run on a personal desktop. By leveraging GPU-based GNN emulators, computationally intensive simulations of ice dynamics and calving can be quickly reproduced on personal desktops without the need for high-performance computing systems. We expect that these types of emulators will facilitate the efficient parameterization of different ice conditions, ice properties, and external climate forcings.

However, although GNNs can successfully replicate the finite-element structure of ISSM simulations and reduce the computational time by leveraging GPUs, it is important to consider the upstream costs for training deep learning emulators to assess their whole-process efficiencies. Table 2 presents the number of learnable parameters and model training time for the GCN, GAT, and EGCN. This training time is recorded from the 500-epoch training of each model on a multiple-GPU system equipped with 8 NVIDIA RTX A5000 GPUs. The GCN takes the least training time, followed by

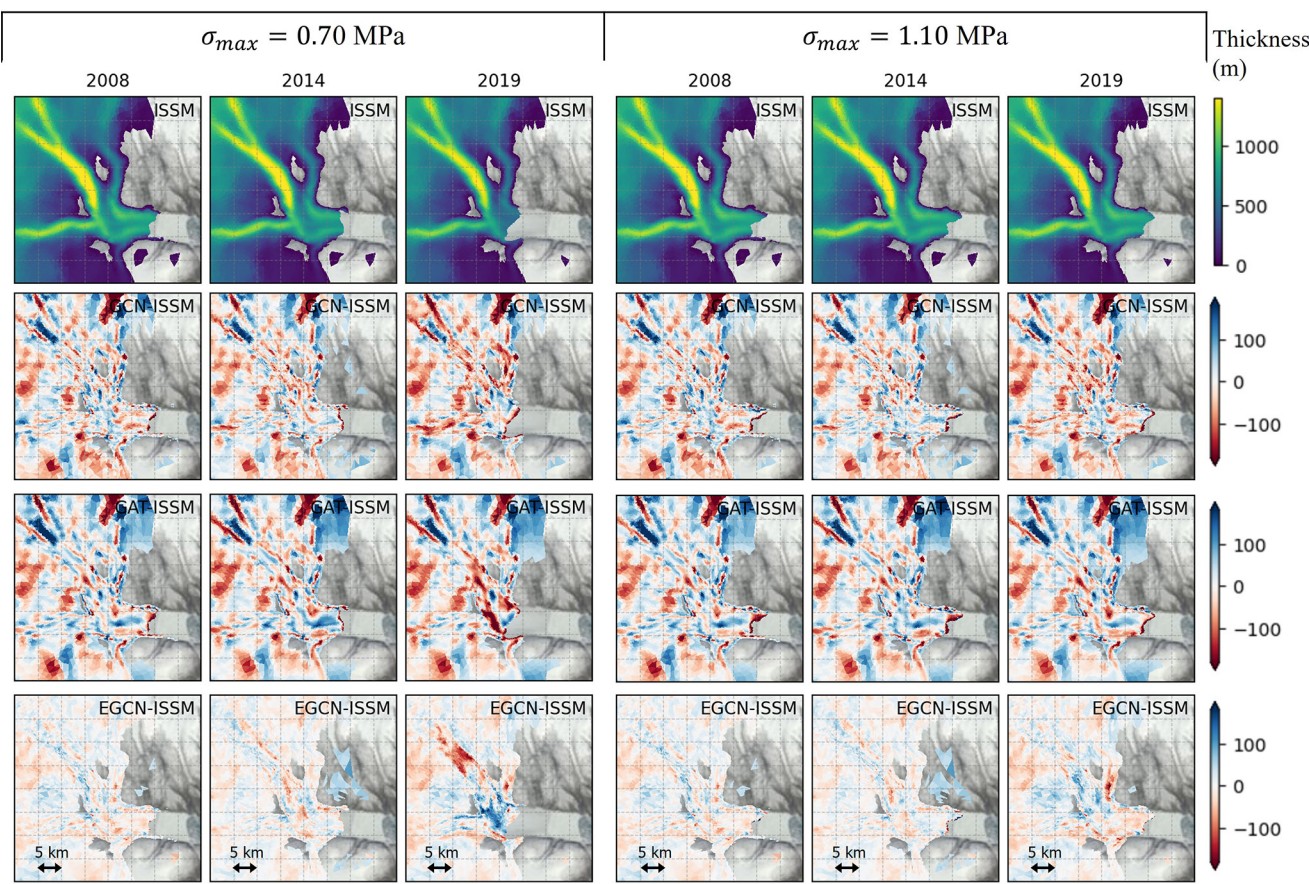

**Figure 5.** Maps of ISSM-simulated ice thickness and difference in ice thickness between GNN emulators and ISSM simulations (GCN, GAT, and EGCN from top to bottom) for 0.70 and 1.10 MPa test $\sigma_{\max}$ values of set A.

EGCN and GAT. Herein, these training times are based on the training datasets of set A; however, the training times of four different sets (Table 1) are similar to each other because they have the same number of training samples. It should also be noted that collecting various ISSM numerical simulations with various $\sigma_{\max}$ values is required to train GNN emulators: in our case, approximately 8500 s ($\sim 2$ h 22 min) are required to collect all 9 simulation results with different $\sigma_{\max}$ values. Nevertheless, given that the emulators show sufficient performances only with 3 training $\sigma_{\max}$ values, fewer simulations would be sufficient for training. Furthermore, we emphasize that these GNN emulators can be efficiently applied to this target glacier with a short implementation time once they are trained, which can accelerate the search for optimal parameterization settings that align the numerical model with observations.

## 5.3  Comparison with real observations

We use the EGCN emulator, trained with set A training samples, to determine the appropriate $\sigma_{\max}$ values that accurately capture the migration of Helheim's calving front, as

the EGCN shows the best performances in predicting ice velocity, ice thickness, and calving front positions (Fig. 3). The satellite-derived ice velocity and ice front observations at each time step are used as input for the EGCN emulator to predict the ice front at the next time step. Then, we identify the optimal $\sigma_{\max}$ value for each time step that minimizes BE of ice front location between prediction and observation (Eq. 13). Given that previous studies (Morlighem et al., 2016; Wilner et al., 2023; Choi et al., 2018) suggest that $\sigma_{\max}$ should fall between 0.7 to 1.1 MPa, we vary $\sigma_{\max}$ incrementally by 0.01 MPa within this range and determine the $\sigma_{\max}$ value with the lowest BE.

Figure 6 shows the temporal variation of optimal $\sigma_{\max}$ derived from the EGCN spanning from 2007 to 2020. Interestingly, there is a significant temporal correlation between the optimal $\sigma_{\max}$ and the terminus positions with coefficient $> 0.5$ ($p$ value $< 0.01$). This result indicates that $\sigma_{\max}$ is an important parameter that determines the calving front migration and should be tuned carefully in calving models. However, we should also note that the temporal variations of optimal $\sigma_{\max}$ and observed calving front do not agree well during the summers of 2017 and 2019. Given that

these two periods coincide with prolonged plume–polynya events lasting more than 20 d in the Helheim Glacier (Melton et al., 2022), we conjecture that buoyant subglacial discharge plumes may have influenced the calving mechanisms during these periods, introducing additional uncertainties to the VM calving law (Everett et al., 2021).

Year-to-year migrations of ice front from observations and numerical models with different $\sigma_{max}$ are shown in Fig. 7. It should be noted that a temporarily constant $\sigma_{max}$ does not represent the observed calving front migration well. If the $\sigma_{max}$ value is set too high ($> 0.8$ MPa), the calving front does not shift or slightly advances after 2017; on the contrary, if $\sigma_{max}$ is set too low ($\leq 0.8$ MPa), the calving front retreats too much such that the modeling results no longer match the observations. On the other hand, the temporally varying optimal $\sigma_{max}$ reproduces the long-term trend of calving front movement along the flow line 1 better than constant $\sigma_{max}$ settings, although the short-term variations do not perfectly match (Fig. 8a). However, we should also note that the calving front migration of flow line 2 is not as well predicted as flow line 1 (Fig. 8b). Although some small advance or retreat along flow line 2 is predicted by numerical simulations with the optimal $\sigma_{max}$ settings, the calving front remains almost the same for 13 years of simulation. This difference between flow line 1 and 2 suggests the need for spatially varying $\sigma_{max}$ parameterization in addition to the temporally varying settings, which is discussed in detail in Sect. 6.

## 6 Discussion

In this section, we discuss the scientific implications of $\sigma_{max}$ variation and the significant role of GNN emulators in improving the physical modeling of glacier calving. Our results reveal that the fine-tuning of temporally varying $\sigma_{max}$ leads to better performance in calving prediction compared to traditionally used constant $\sigma_{max}$ parameterization. As a parameter indicating how difficult it is for calving to happen, a greater $\sigma_{max}$ means a lower rate of calving, and a lower $\sigma_{max}$ means a higher rate of calving. Previous studies have shown that the physical condition of mélange around the ice front, such as ice thickness and concentration, changes the frequency of calving events; a weaker or disappearing mélange area leads to an increase in calving events, which corresponds to lower $\sigma_{max}$ (Wehrlé et al., 2023; Xie et al., 2019; Meng et al., 2025). Although this study is focused on finding optimal $\sigma_{max}$ values using GNNs, exploring the linkage between optimal $\sigma_{max}$ and mélange conditions can contribute to the accurate parameterization of $\sigma_{max}$ for calving models. Additionally, given that mélange and sea ice conditions can vary along the ice front, assigning spatially varying $\sigma_{max}$ can address the discrepancies in calving front migration for two flow lines shown in Figs. 7 and 8. To set spatially varying $\sigma_{max}$, we can consider finding the best $\sigma_{max}$ for multiple flow lines and interpolate these values at the ice front. It

would also be valuable to investigate how $\sigma_{max}$ interacts with other environmental factors, including tidal forcing and deep water circulation, which are known as significant drivers for calving events (O'Neel et al., 2003; Bassis and Jacobs, 2013; Slater and Straneo, 2022). Consequently, finding the optimal calving parameterization setting and connecting it to external environmental drivers will contribute to the accurate modeling of glacier mass loss and sea level rise beyond the conventional constant calving calibration.

Given the scientific implication of temporally varying $\sigma_{max}$, the significance of this study lies in the (i) fidelity and (ii) computational efficiency of using GNN architectures, especially EGCN, in emulating numerical ice sheet models operating on unstructured meshes to find the optimal calving parameterization. By applying GNNs directly on raw unstructured meshes, we can keep high resolution around the fast ice area to delineate the calving front accurately. In particular, EGCN shows the best fidelity in predicting ice thickness due to its equivariance concept throughout the graph structures. Moreover, the use of GNN emulators results in a dramatic increase in computational efficiency, with speeds 30–34 times faster than the ISSM simulations. This speed-up allows us to find the temporally varying optimal $\sigma_{max}$ values quickly. Given that running numerical simulations every time to determine the optimal $\sigma_{max}$ setting is extremely time-consuming, our fast GNN emulators reduce the workload associated with calving calibration.

Besides these advantages of GNN architectures, we emphasize that GNN emulators have significant potential for further improvement, particularly through integration with various architectures, including recurrent neural networks (RNNs) (Wu et al., 2021). For example, while our GNN emulator only implements a mapping between a previous time step ($t-1$) to the next time step ($t$), embedding recurrent units into the GNN architecture can make it possible to find the sequential relationships between ice sheet dynamics and calving parameters (i.e., mapping between multiple time steps). This recurrent GNN architecture will be able to predict how the historical context of calving parameters affects ice dynamics and ice front migration.

However, it is also worth noting several limitations of our approach. Firstly, despite the computational efficiency of GNN emulators, they should be trained using numerical simulations. Since the model performance is highly dependent on the quality of training datasets, simulation data should be collected carefully with appropriate parameters. Second, the collection of training datasets from various climatological scenarios can be helpful for better generalizability and reliability of emulators to predict future glacier behaviors; however, this process can be exceptionally time-consuming. This generalizability issue can also be associated with the applicability of our GNN emulators for other glaciers beyond Helheim Glacier. To assess the fidelity and transferability of our GNN emulators trained on Helheim Glacier, we apply them to the Pine Island Glacier (PIG), Antarctica, which has

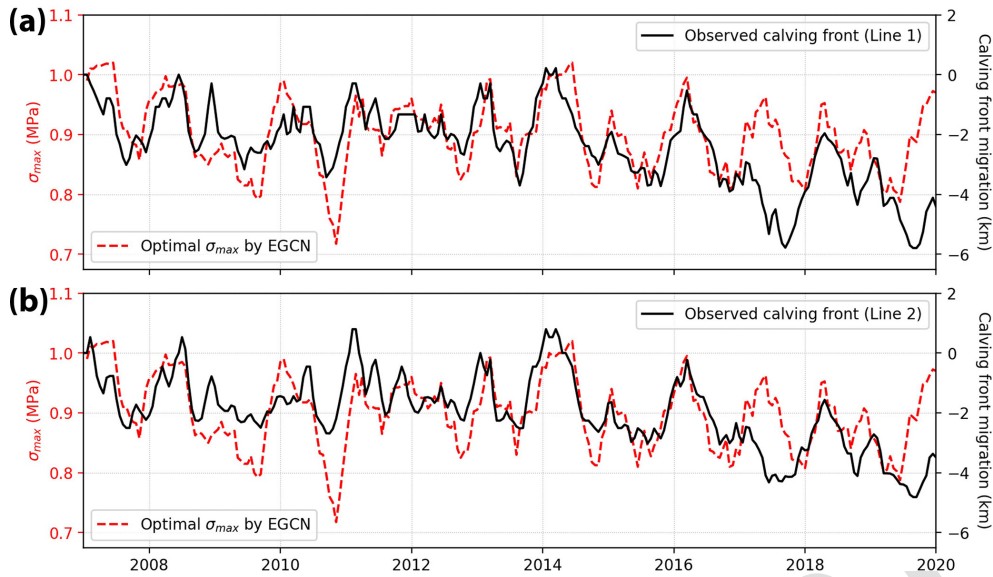

**Figure 6.** Temporal variations of optimal $\sigma_{max}$ derived by EGCN (red dashed line) and the temporal movement of terminus of flow line 1 **(a)** and flow line 2 **(b)** in Fig. 1b.

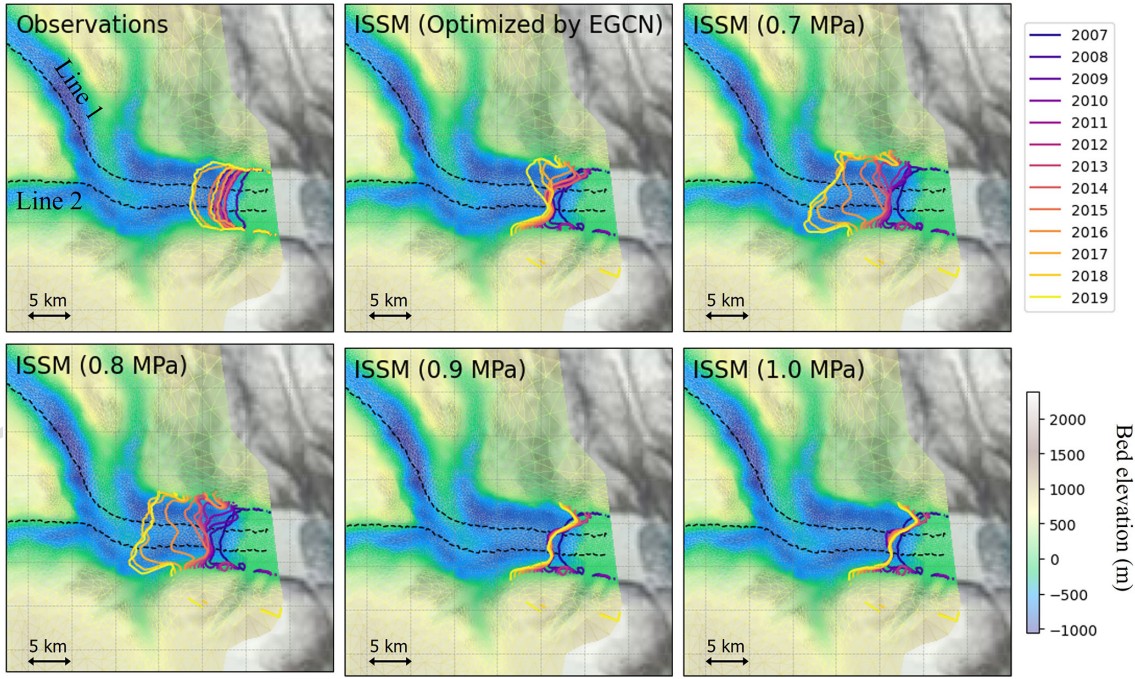

**Figure 7.** Annual movement of ice front observed by satellite imagery and modeled by ISSM with different $\sigma_{max}$ parameterizations: optimal $\sigma_{max}$ values from EGCN and constant $\sigma_{max}$ values (0.7, 0.8, 0.9, and 1.0 MPa).

different ice velocity and thickness distributions. The ice velocity RMSE in PIG ranges from 200 to 400 m a$^{-1}$ (2–4 times greater RMSE than Helheim Glacier), and the ice thickness RMSE ranges from 120 to 200 m (5–10 times greater RMSE than Helheim Glacier) (Figs. S1, S2, and Table S1 in the Supplement). These results suggest the importance of training GNN emulators on datasets encompassing a diverse range

of ice velocity and thickness distributions to ensure broad generalizability and applicability across different glacier settings. Finally, although we employ the VM calving law to determine calving front migration, it is important to recognize that calving mechanisms are not yet fully understood and may be more complex than those represented by the VM law. While the VM method has been validated for many glaciers

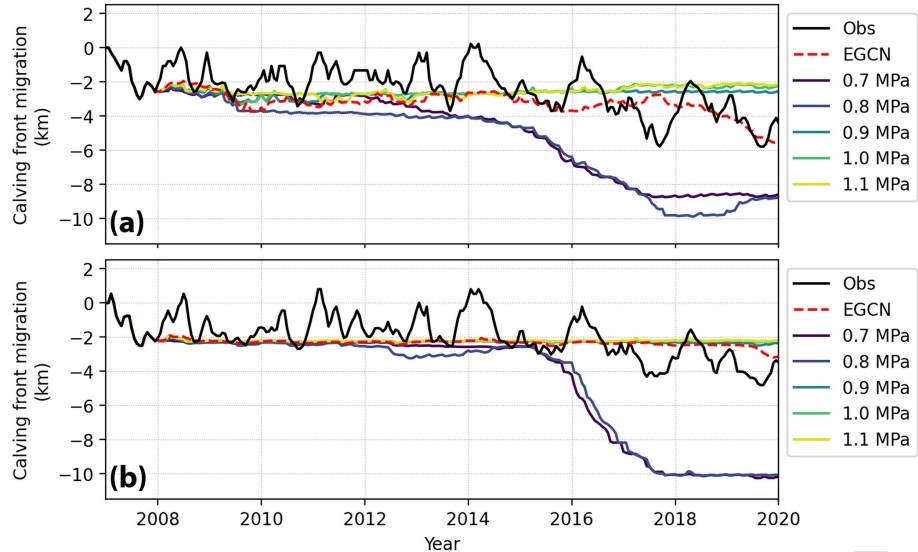

**Figure 8.** Movement of ice front location for **(a)** flow line 1 and **(b)** flow line 2 predicted by numerical models with different $\sigma_{max}$ parameterizations; optimal $\sigma_{max}$ values from EGCN and constant $\sigma_{max}$ values (0.7, 0.8, 0.9, 1.0, and 1.1 MPa). The observed ice front movement is drawn as a black solid line.

in Greenland and Antarctica (Choi et al., 2018; Wilner et al., 2023), detailed calving mechanisms remain elusive. Thus, although our GNN emulators can provide valuable insights into how to select appropriate $\sigma_{max}$ for VM calving law, they rely on our current imperfect physical understanding of calving.

## 7 Conclusions

This study develops three graph neural networks – GCN, GAT, and EGCN – as surrogate models to reproduce finite-element ice dynamics and calving retrieved from the ISSM. After training these GNNs with 13-year transient simulations of Helheim Glacier, they demonstrate significant spatiotemporal agreement with ISSM simulations in predicting ice velocity, ice thickness, and ice front location. The GNN emulators successfully reproduce the retreat of the ice front for a lower calving stress threshold $\sigma_{max}$, as well as the stable condition of the ice front for a higher $\sigma_{max}$. Additionally, the GNN emulators show only $< 10\,\%$–$20\,\%$ of uncertainties in predicting out-of-sample $\sigma_{max}$ values, implying a better capability in extrapolation and generalizability than previous statistical emulators. Among the three GNN architectures, the EGCN shows the best robustness in the prediction of ice thickness and velocity by preserving the equivariance of graph structures. By leveraging 30–34 times faster computational time of the GPU-based GNN emulators compared to numerical simulations, we efficiently find temporally varying optimal $\sigma_{max}$ parameterization for the von Mises calving law. By applying satellite-derived observations to the EGCN, we find that the temporal variations of optimal $\sigma_{max}$ have

a significant correlation with the retreat and advance of the calving front from 2007 to 2020. Additionally, the numerical simulations based on the fine-tuned $\sigma_{max}$ calibration show a better agreement with observations compared to conventional constant $\sigma_{max}$ setting. Therefore, this result highlights the importance of setting $\sigma_{max}$ values appropriately to improve the reliability of numerical models. As the first attempt to use GNNs for calibrating calving parameters, GNN emulators can further contribute to improving the prediction accuracy of ice sheet mass loss and resulting sea level rise.

## Appendix A: *T* tests for training and test datasets

We conduct statistical *t* tests to examine (i) whether test error is significantly different from training error (Table A1) and (ii) whether a certain GNN architecture shows significantly better performance than others (Table A2).

**Table A1.** Reduction in test error relative to training error in percentage (%) for each training/test set and GNN emulator. The positive values mean lower error in test datasets than in training datasets. If the mean difference is not statistically significant (*p* value > 0.01; i.e., statistically they have the same mean), we highlight it in bold.

| Set | Ice velocity RMSE | | | Ice thickness RMSE | | | BE | | |
|-----|------|------|------|------|------|------|------|------|------|
|     | GCN | GAT | EGCN | GCN | GAT | EGCN | GCN | GAT | EGCN |
| A | **0.2** | **−0.2** | **0.1** | **0.0** | **−0.1** | **0.1** | **2.9** | **0.0** | **0.0** |
| B | 7.8 | 4.9 | 7.0 | −0.8 | −0.4 | 3.8 | −10.7 | −22.7 | −7.1 |
| C | −0.5 | −0.8 | −2.9 | −0.4 | −0.2 | −1.2 | −11.8 TS4 | −15.1 | −6.7 |
| D | −1.3 | −2.3 | −24.3 | −1.7 | −0.5 | −2.5 | −12.5 | −16.7 | −17.2 |

**Table A2.** Performance improvement by GAT and EGCN relative to GCN (i.e., error reduction relative to GCN error) in percentage (%). The positive values mean a better performance by GAT or EGCN compared to GCN. If the mean difference between the two models is not statistically significant (*p* value > 0.01; i.e., statistically they have the same mean), we highlight it in bold.

| Set | Ice velocity | | Ice thickness | | Calving front | |
|-----|------|------|------|------|------|------|
|     | GAT | EGCN | GAT | EGCN | GAT | EGCN |
| A | 24.9 | 78.9 | −2.7 | 50.3 | 11.8 | 11.8 |
| B | 33.3 | 77.7 | 4.8 | 51.4 | 12.9 | **3.2** |
| C | 32.3 | 80.4 | **−2.2** | 46.5 | **0.0** | 15.8 |
| D | 32.1 | 77.7 | 4.3 | 46.1 | **2.8** | **5.5** |

*Code and data availability.* All the code and data are available at https://github.com/BinaLab/ISSM_GNN (last access: 9 July 2025) and https://doi.org/10.5281/zenodo.11392220 (Koo, 2024).

*Supplement.* The supplement related to this article is available online at [the link will be implemented upon publication].

*Author contributions.* YK: formal analysis, investigation, methodology, validation, visualization, writing (original draft preparation); GC: data curation, resources, software, writing (review and editing); MM: resources, software, writing (review and editing); MR: conceptualization, methodology, resources, funding acquisition, project administration, supervision, writing (review and editing).

*Competing interests.* At least one of the (co-)authors is a member of the editorial board of *The Cryosphere*. The peer-review process was guided by an independent editor, and the authors also have no other competing interests to declare.

ther geographical representation in this paper. While Copernicus Publications makes every effort to include appropriate place names, the final responsibility lies with the authors.

*Acknowledgements.* This work is supported by NSF BIGDATA (IIS-1838230, 2308649) and NSF Leadership Class Computing (OAC-2139536) awards. Gong Cheng and Mathieu Morlighem were supported by the Heising Simons Foundation grant nos. 2019-1161 and 2021-3059.

*Financial support.* This research has been supported by the National Science Foundation (grant nos. 1838230, 2308649, and 2139536) and the Heising-Simons Foundation (grant nos. 2019-1161 and 2021-3059). TS5

*Review statement.* This paper was edited by Johannes J. Fürst and reviewed by three anonymous referees.

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

**Remarks from the typesetter**

TS1    Please give an explanation of why this needs to be changed. We have to ask the handling editor for approval. Thanks.

TS2    Please give an explanation of why this needs to be changed. We have to ask the handling editor for approval. Thanks.

TS3    Please give an explanation of why this needs to be changed. We have to ask the handling editor for approval. Thanks.

TS4    Please confirm "$-11.8$".

TS5    All funders and grant numbers related to the study must be mentioned in the Financial support section and may be repeated in the Acknowledgements. I would kindly ask you to check if the funding information in the Financial support section is correct and complete. The content of both section should not differ if the funding information is kept in the Acknowledgements.

TS6    Please confirm adjusted information.