# Peer review of "Calibrating calving parameterizations using graph neural network emulators: Application to Helheim Glacier, East Greenland"

_EGUsphere, 2024_

## Referee Comment (RC1)

**Review of 'Calibrating calving parameterizations using graph neural network emulators' by Koo et al.**

**July 2024**

In this manuscript, Koo et al. describe the application of several variants of a graph neural network to emulate ISSM at Helheim Glacier. I am unsure as to exactly what this emulator does - whether it is similar to IGM in producing an approximate solution operator to the (in this case coupled) momentum and mass conservation equations or whether it is more similar to He (2023) in being geometry specific - but in general terms it is trained to reproduce the predicted ice velocity and thickness as a function of time and space.

I think that this paper has the potential to be a useful contribution to the literature, and the goal of coming up with emulators that operate naturally in the same discrete setting as FEM-based ice sheet models is worthy. Additionally, the saliency with which the neural networks either learn or memorize (I'm not entirely sure which) the model's behavior is impressive. However the manuscript needs significant clarification (and in some cases moderation) of its claims in order to assess their veracity and utility. In particular, the methods are unclear and not reproducible - as mentioned above, I am not clear what the features used for prediction actually are. Additionally, the paper does not include a fair representation of the computational costs of the proposed methods. Finally, the paper makes many claims that its proposed methodology is better than others without providing sufficient evidence to back up that claim.

Detailed comments are below.

**L63** Downs (2023), which is already referenced in this paper, provides significant insight into this question, and also serves as another example of a surrogate model being used to infer calving dynamics at Helheim Glacier (though not a GNN).

**L75** The acronym VM should be defined here.

**L78** Physically, why should the stress threshold have to be calibrated on a glacier-by-glacier basis?

**Sec. 2.3** This section should include some earlier literature. It would be worth looking at Tarasov et al. (2012, https://doi.org/10.1016/j.epsl.2011.09.010) and Brinkerhoff et al. (2021, https://doi.org/10.1017/jog.2020.112).

**L123** I don't understand why GNN's would be particularly suited for ice front migration relative to other architectures.

**L146** 'adjacent' → 'adjacency'.

**L149** Finite elements are, at their core, interpolants. Does bilinear interpolation here just mean that the FEM solution is evaluated at grid points? Or is some other interpolant introduced?

**L179** the square root does not need to be defined.

**L180** 'of' → 'with'.

**L185-187** I don't think that this description makes sense. The architecture *for some particular hidden layer* weights different adjacent nodes differently for an operation that is otherwise the same as vanilla graph convolution. The resulting convolutions are then stacked – graph convolution is stacked, but the attention operation is internal to that.

**Eq. 6** I am not sure whether this equation is correct, but I am also not sure whether it's necessary - it seems like maybe something very detailed that appears in the reference (although it does seem weird to concatenate the projected feature vectors like this, which would seem to make the attention scores sensitive to the order of arguments). Maybe okay to forego?

**Fig. 3** I don't think that this figure is helpful for illustrating how either of these models work.

**Sec. 4.3** This section is really difficult to understand. One thing that I can glean from this is that this is operation is $O(n^2)$ in the number of graph nodes - does that have any implications for performance?

**Sec. 4.5** This description of the model's inputs and outputs should appear at the beginning of the methods section. Furthermore, specifically what these features mean needs to be much more clearly described - as it stands, I cannot assess the quality of this work because, despite looking at both the manuscript and the linked code, I cannot tell what this emulator is building a mapping between. There are a few things implications to mention associated with this.

First, it appears that the time $t$ is explicitly included as a feature. This then implies that the surrogate is *not* time-invariant and the mapping should be thought of as a tool for downstream analysis (like He (2023) or Downs(2023)) rather than as learning the solution operator for Stokes' equations (or Stokes plus continuity since the present work claims to predict thickness as well). This is fine - such models can be very useful - but it mandates a change in language to reflect the fact that it is unlikely that this method can generalize to other locations or times.

Second, the velocity components at some previous time (it's not clear whether this is from the model's previous time step or at the beginning of the simulation - this notation needs to be modified to be more clear) is used as a feature. This is an unfortunate choice because one thing that we know about ice physics is that the velocity is approximately diagnostic of the geometry - if you know the latter, you can predict the former. In the absence of that property, how can this model be started? Is it the case that ISSM has to be run first before this emulator can be applied?

What is a 'forwarding process'? What is a graph 'structure'? Is this just the collection of node/edge features?

This section is essential to understanding what is going on (more essential even than architecture choice), yet it's only two paragraphs long and does not have sufficient information to allow for reproducibility.

**Eqs. 11, 12, 13** These are all standard definitions that do not need to be included here.

**L269** 'out of sample' is perhaps a bit of an overstatement - the degree of correlation between neighboring $\sigma$ values would very likely be quite high - as a check, it would be interesting to see what error is induced by comparing model predictions made using $\sigma_{max} = 0.8$ to the withheld test set values for $\sigma_{max} = 0.75$ (or something like that). I expect the metrics would be similar because there is little difference between such small variations in the parameter. A more useful test would be to train on just the two extremal values (0.7,1.1) and see if it can still interpolate well.

**Table 1** These metrics are all so inflated as to be useless. Is it possible to come up with relative metrics that use more significant digits? I also think it would be better to combine Tables 2 and 3 with Table 1 - it is useful to think of model accuracy relative to model size and expense.

**Sec. 5.1** A single study does not establish the superiority of GNNs over CNNs for tasks such as these - it could (and very likely is) the case that the current results are incidental (or cherry picked) and that different researchers could find different conclusions. Furthermore, with respect to efficiency, there are many tricks that can be performed on CNNs to make them faster that have no analogue for an unstructured mesh, none of which were presumably included in the present analysis. The style of NeurIPS or similar notwithstanding, it is generally unhelpful to try to establish the primacy of one method over another in this way, and I would encourage the authors to either undertake a much more controlled and systematic comparison between methods or to reframe this as less of a competition.

**Sec. 5.2** It is frankly absurd to not include even a mention of the computational cost associated with increasing the training data, which - so far as I can tell - must be repeated for any new geometry or parameter or location. Ignoring this cost does of course lead to much more impressive speed-ups, but these are not real. I would expect this problem to become considerably more severe when trying to use this technique to emulate models that are a function of more than a single parameter - the curse of dimensionality still applies. Furthermore, the notion that a GNN will be more suitable than a CNN for higher resolution modelling is another strawman because it ignores the fact that generating the cost of generating the training data (which is presumably more expensive than any network training) is also going to scale proportionaly with resolution. I would encourage the authors to revisit this entire section with a more sober perspective aimed at delivering a factual assessment of the present work's utility.

**L351** This is only true if the model's behavior in response to variations in this new parameter is as well-quantified as it is to the parameter considered in this work, which may not be true, or may not be tractable.

**Sec. 6** Again, I strongly urge the authors not to try to cast their work in terms of 'superiority' - the present work does not provide sufficient evidence for such blanket statements, nor is it necessary.

**Fig. 7** This needs a more descriptive caption - I am struggling to see what these figures are showing.

**L370–371** Is this statement really necessary?

---

## Author Comment (AC1)

**Reviewer 1**

In this manuscript, Koo et al. describe the application of several variants of a graph neural network to emulate ISSM at Helheim Glacier. I am unsure as to exactly what this emulator does- whether it is similar to IGM in producing an approximate solution operator to the (in this case coupled) momentum and mass conservation equations or whether it is more similar to He (2023) in being geometry specific- but in general terms it is trained to reproduce the predicted ice velocity and thickness as a function of time and space.

I think that this paper has the potential to be a useful contribution to the literature, and the goal of coming up with emulators that operate naturally in the same discrete setting as FEM-based ice sheet models is worthy. Additionally, the saliency with which the neural networks either learn or memorize (I'm not entirely sure which) the model's behavior is impressive. However, the manuscript needs significant clarification (and in some cases moderation) of its claims in order to assess their veracity and utility. In particular, the methods are unclear and not reproducible- as mentioned above, I am not clear what the features used for prediction actually are. Additionally, the paper does not include a fair representation of the computational costs of the proposed methods. Finally, the paper makes many claims that its proposed methodology is better than others without providing sufficient evidence to back up that claim. Detailed comments are below.

- Thank you for your constructive comments. We review your comments carefully and will make necessary changes accordingly.

L63 Downs (2023), which is already referenced in this paper, provides significant insight into this question, and also serves as another example of a surrogate model being used to infer calving dynamics at Helheim Glacier (though not a GNN).

- We will add brief explanations about how Downs et al. 2023 inferred the calving dynamics in Helheim Glacier, and how our approach differs (although we consider our application as an illustrative example).

L75 The acronym VM should be defined here.

- VM is already defined in L29.

L78 Physically, why should the stress threshold have to be calibrated on a glacier-by-glacier basis?

- This is a good point that is a bit out of the scope of this paper. Previous studies (e.g. Morlighem et al. 2019) found that most glaciers in Northwest Greenland had a threshold around 1 MPa (see the first paragraph of their results section). But they also found that some glaciers "needed" a different threshold, even though it is not fully clear why that is the case. Some of these variations could be due to a bias in the model initialization or the

forcing, but also due to different levels of damage of individual glaciers, which is not necessarily captured by the model. We will add a short explanation to the text.

Sec. 2.3 This section should include some earlier literature. It would be worth looking at Tarasov et al. (2012, https://doi.org/10.1016/j.epsl.2011.09.010) and Brinkerhoff et al. (2021, https://doi.org/10.1017/jog.2020.112).

- We will add Tarasov et al. 2012 and Brinkerhoff et al. 2021 to Section 2.3.

L123 I don't understand why GNN's would be particularly suited for ice front migration relative to other architectures.

- We apologize for missing a detailed explanation. We meant that this was particularly attractive for finite element ice sheet models because of how they typically discretize the model domain. ISSM is a numerical ice sheet model that relies on unstructured meshes: it uses a finer resolution in the fast ice region and a coarser resolution in the slow ice region to optimize computational efficiency. However, CNNs inherently rely on a fixed resolution (regular grid) for all points, which may not capture the detailed resolution at the ice front. If we want a sufficiently fine resolution at the ice front, the grid size of the CNN would be very fine, which increases the computational demand exponentially. On the contrary, since GNNs directly use the ISSM unstructured meshes, they can keep the advantages of the ISSM numerical simulations in a more natural way. Hence, GNNs can be a better option to (1) obtain accurate ice front mitigation by embedding the interaction between neighboring nodes, (2) obtain a sufficiently fine resolution to capture ice front migration, (3) minimize the computational load. We will add a detailed explanation of why we chose GNN as the backbone architecture.

L146 'adjacent' → 'adjacency'.

- Done.

L149 Finite elements are, at their core, interpolants. Does bilinear interpolation here just mean that the FEM solution is evaluated at grid points? Or is some other interpolant introduced?

- Yes. Since the FEM solution is not provided as regular grid points, we calculate the velocity and thickness solutions for regular grid points from the FEM results.

L179 the square root does not need to be defined.

- Done.

L180 'of' → 'with'.

- Done.

L185-187 I don't think that this description makes sense. The architecture *for some particular hidden layer* weights different adjacent nodes differently for an operation that is otherwise the same as vanilla graph convolution. The resulting convolutions are then stacked– graph convolution is stacked, but the attention operation is internal to that.

- Correct. This self-attention operation is internal to a hidden layer. We will add some clarification about it.

Eq. 6: I am not sure whether this equation is correct, but I am also not sure whether it's necessary- it seems like maybe something very detailed that appears in the reference (although it does seem weird to concatenate the projected feature vectors like this, which would seem to make the attention scores sensitive to the order of arguments). Maybe okay to forego?

- We used the same equation from the reference paper (Velickovic, 2018), but we remove this equation from the revised manuscript because it is just a detailed version of Eq. 5.

Fig. 3 I don't think that this figure is helpful for illustrating how either of these models work.

- This figure was intended to illustrate the concept of attention and equivariance. We remove this figure from the revised manuscript.

Sec. 4.3 This section is really difficult to understand. One thing that I can glean from this is that this is operation is O(n2) in the number of graph nodes- does that have any implications for performance?

- We add more explanation about the EGCN model in Section 4.3. We cannot explain every detail of how the EGCN works, but the details about the EGCN, including how this architecture guarantees equivariance, can be found in Satorras et al., (2021).
- Technically speaking, EGCN is an operation of $O(n^2)$ because it operates on all graph nodes, while the GCN and GAT operate only on the adjacent nodes. That is, the EGCN operates on all graph nodes to preserve equivariance in the entire graph. Thus, the EGCN requires more processing time than the GCN and GAT, as shown in Section 5.2. We add more discussion about this in Section 5.2.

Sec. 4.5 This description of the model's inputs and outputs should appear at the beginning of the methods section. Furthermore, specifically what these features mean needs to be much more clearly described- as it stands, I cannot assess the quality of this work because, despite looking at both the manuscript and the linked code, I cannot tell what this emulator is building a mapping between. There are a few things implications to mention associated with this. First, it appears that the time t is explicitly included as a feature. This then implies that the surrogate is not time-invariant and the mapping should be thought of as a tool for downstream analysis (like He (2023) or Downs(2023)) rather than as learning the solution operator for Stokes' equations (or Stokes plus continuity since the present work claims to predict thickness as well). This is finesuch models can be very useful- but it mandates a change in language to reflect the fact that it is unlikely that this method can generalize to other locations or times. Second, the velocity components at some previous time (it's not clear whether this is from the model's previous time step or at the beginning of the simulation- this notation needs to be modified to be more clear) is used as a feature. This is an unfortunate choice because one thing that we know about ice physics is that the velocity is approximately diagnostic of the geometry- if you know the latter, you can predict the former. In the absence of that property, how can this model be started? Is it the case that ISSM has to be run first before this emulator can be applied? What is a 'forwarding process'? What is a graph 'structure'? Is this just the collection of node/edge features? This section is essential to understanding what is going on (more essential even than architecture choice), yet it's only two paragraphs long and does not have sufficient information to allow for reproducibility.

- Thank you for your detailed comment. We will add more descriptions about the input datasets at the beginning of the Method section. Regarding the mapping of input and output features, Fig. 2 helps to understand how this emulator works between input and output features.
- Regarding your first point about the time-invariant, we agree that our emulator is not time-invariant. There are a few reasons why we include time as an input feature. We would like to make our emulator provide the variables that regular measurements are available for. This facilitates the comparison between emulator output and real observations and finding good parameterizations. Therefore, we select velocity and ice front, whose observations are available in real-time, as the output features of our emulators. In our emulator, it was necessary to include time as an input feature to indicate the temporal evolution of ice thickness. If real-time ice thickness observations were available, we would include ice thickness as both input and output features, exclude time from the input features, and make our emulator 100 % time-invariant. Unfortunately, however, real-time ice thickness observations are not available. Thus, we could not use real-time "previous-time-step" ice thickness as an input feature to predict "next-time-step" ice thickness. Instead, we include time and initial ice thickness as input features so that the emulator can indirectly see the impacts of the temporal evolution of ice thickness at a specific time step. We will add this explanation about input feature selection to Section 4.5 and mention that our method can have limitations when generalized to other locations or times.
- In this study, we would like to find the parameterization that matches best real observations into numerical models. Therefore, we design our emulator to predict the "next-time-step" velocity from the previous-time-step velocity with certain parameterization settings, assuming that the parameterization changes the ice velocity. This parameterization includes the calving threshold (sigma_max) and the geometry of the ice sheet as well, which is included as the input feature of the emulator (i.e., bed elevation and initial ice thickness).

- Here, the forwarding process just indicates the determination of output from input features via a neural network. 'Graph structure' is the collection of nodes and edges. We will clarify the meaning of these terms in the revised manuscript.

Eqs. 11, 12, 13 These are all standard definitions that do not need to be included here.

- Although the Eqs. 11-13 are standard definitions, we would like to keep them for completeness because some readers may wonder how these metrics are calculated.

L269 'out of sample' is perhaps a bit of an overstatement- the degree of correlation between neighboring σ values would very likely be quite high- as a check, it would be interesting to see what error is induced by comparing model predictions made using σmax = 0.8 to the withheld test set values for σmax = 0.75 (or something like that). I expect the metrics would be similar because there is little difference between such small variations in the parameter. A more useful test would be to train on just the two extremal values (0.7,1.1) and see if it can still interpolate well.

- We agree that the degree of correlation between neighboring σ values can be high. In the Appendix, we will add some additional experiments with neighboring σ values: comparing σmax = 0.8 and 0.7 to the withheld test set values for σmax = 0.75, and comparing σmax = 0.85 and 0.95 to σmax = 0.90.

Table 1 These metrics are all so inflated as to be useless. Is it possible to come up with relative metrics that use more significant digits? I also think it would be better to combine Tables 2 and 3 with Table 1- it is useful to think of model accuracy relative to model size and expense.

- We found that the R values are too inflated because we calculated this metric for the entire glacier domain. However, the significant differences are only near the ice front and ice stream, while the other regions with slow ice show insignificant differences. We will recalculate these metrics only near the ice front and ice stream.
- Although combining Tables 1, 2, and 3 would be useful to see the model accuracy relative to the model size and expense, we are concerned that the combined table is too distracting because it includes too much information. Instead, we will combine Tables 2 and 3, which will make a new Table 2, so that the readers can focus on the model fidelity in Table 1 and computational expenses in Table 2.

Sec. 5.1 A single study does not establish the superiority of GNNs over CNNs for tasks such as these- it could (and very likely is) the case that the current results are incidental (or cherry picked) and that different researchers could find different conclusions. Furthermore, with respect to efficiency, there are many tricks that can be performed on CNNs to make them faster that have no analogue for an unstructured mesh, none of which were presumably included in the present analysis. The style of NeurIPS or similar notwithstanding, it is generally unhelpful to try to establish the primacy of one method over another in this way, and I would encourage the authors

to either undertake a much more controlled and systematic comparison between methods or to reframe this as less of a competition.

- This is a good point, but we still argue that the advantage of using GNN is that it can directly use the unstructured meshes of ISSM. The unstructured meshes of ISSM (and other finite-element ice sheet models) are characterized by variable resolutions, which allocate high resolution to fast ice region and low resolution to slow ice region. While GNNs can directly use the finite-element mesh data structures, CNNs cannot do it as they only operate on regular grids. Since ISSM uses unstructured meshes, using CNNs as emulators for ISSM can introduce two problems: (1) the CNN grid with fixed resolution can lose dynamical details in fast ice areas; (2) the CNN grid requires unnecessary computational demands in slow ice areas. We submitted another research paper to another journal that applies a similar GCN architecture to another glacier in Antarctica. In this paper, we include more detailed discussions about the advantages of using GCN over CNN in terms of model fidelity and computational efficiency with various spatial resolution conditions. The preprint version of our paper is available at this link: https://arxiv.org/abs/2402.05291. In the revised version, we will highlight the advantages of GNN over CNN, particularly for the ISSM ice sheet model with unstructured meshes.
- We agree that several tricks can be applied to CNNs. However, in this study, we would like to just compare the most basic architectures. The GCN, GAT, and EGCN architectures are also very basic architectures designed for non-regular graph structures: GCN uses convolutional operations on graphs; GAT adds attention operation to the graph convolutional layer; EGCN uses equivariance graph convolutional layers to preserve equivariance in graph structures. Given that there are numerous modifications of basic CNN architectures, handling all of them is beyond the scope of this research paper. Nevertheless, since ISSM operates on unstructured meshes, GNNs have fundamental advantages over CNNs for emulating ISSM. In this paper, we would like to highlight the fundamental limitations of CNN in handling unstructured mesh, especially in delineating the calving front. We argue that GNNs perform better than CNN, at least within the architectures that we have tested, but others can find CNN architectures that may do a better job with exhaustive searches and numerous tricks.

Sec. 5.2 It is frankly absurd to not include even a mention of the computational cost associated with increasing the training data, which- so far as I can tell- must be repeated for any new geometry or parameter or location. Ignoring this cost does of course lead to much more impressive speed-ups, but these are not real. I would expect this problem to become considerably more severe when trying to use this technique to emulate models that are a function of more than a single parameter- the curse of dimensionality still applies. Furthermore, the notion that a GNN will be more suitable than a CNN for higher resolution modelling is another strawman because it ignores the fact that generating the cost of generating the training data (which is presumably more expensive than any network training) is also going to scale proportionally with resolution. I

would encourage the authors to revisit this entire section with a more sober perspective aimed at delivering a factual assessment of the present work's utility.

- We agree that the repeated application of our GNN emulators to another location or parameter would require more computational cost. In this study, our simulation is only focusing on the VM calving law that requires only σmax for parameterization; however, if we use another calving law that requires additional parameters, it would take much more time to collect the simulation data from various parameter settings. Such computational demands with multiple parameters and different locations can be inferred from our results. Additionally, we want to emphasize that the computational significance of deep learning emulators is that they can find the statistical relationships between input and output features and the optimal parameter settings once they are trained from the provided simulation data. We will add a brief mention of this problem in Section 5.2.
- Generating training data is not a big task for GNN. Since GNN directly uses the meshes of ISSM, there is no significantly time-consuming extra workload. The only thing we need to do to generate a training dataset for GNN is to define the graph structures with the ISSM meshes, elements, and adjacency matrix of nodes, and it takes only a few seconds in a local machine. On the contrary, this additional workload is applied to CNNs rather than GNNs because CNNs require to interpolate all the features from the unstructured mesh of ISSM into regular grids.

L351 This is only true if the model's behavior in response to variations in this new parameter is as well quantified as it is to the parameter considered in this work, which may not be true, or may not be tractable.

- Agreed. Since it is not clear how the other parameters, which are not considered in our emulator, have impacts on calving, we will remove this part in the revised version. Instead, we will just briefly mention the limitation of the VM calving model.

Sec. 6 Again, I strongly urge the authors not to try to cast their work in terms of 'superiority'- the present work does not provide sufficient evidence for such blanket statements, nor is it necessary.

- Please see our previous response about the "advantages" of using GNNs over CNNs in replicating ISSM, which uses unstructured meshes, in terms of model fidelity and computational efficiency. We will however tone them down following your comment.

Fig. 7 This needs a more descriptive caption- I am struggling to see what these figures are showing.

- Sorry for the too-short caption. We will add a more descriptive caption for this figure.

L370–371 Is this statement really necessary?

- We will remove this statement in the revised version.

**References**

- Choi, Y., Morlighem, M., Wood, M. & Bondzio, J. H. Comparison of four calving laws to model Greenland outlet glaciers. Cryosphere 12, 3735–3746, https://doi.org/10.5194/tc-12-3735-2018, 2018.
- Wilner, J. A., Morlighem, M. & Cheng, G. Evaluation of four calving laws for Antarctic ice shelves. Cryosphere 17, 4889–4901, 2023.
- Morlighem, M., J. Bondzio, H. Seroussi, E. Rignot, E. Larour, A. Humbert, and S. Rebuffi (2016), Modeling of Store Gletscher's calving dynamics, West Greenland, in response to ocean thermal forcing, Geophys. Res. Lett., 43, 2659–2666, doi:10.1002/2016GL067695.
- Petrovic, J.J. Review Mechanical properties of ice and snow. Journal of Materials Science 38, 1–6 (2003). https://doi.org/10.1023/A:1021134128038
- Velickovic, P., Cucurull, G., Casanova, A., Romero, A., Liò, P., and Bengio, Y.: Graph Attention Networks, https://doi.org/10.48550/arXiv.1710.10903, 2018
- Satorras, V.G., Hoogeboom, E. and Welling, M., 2021, July. E (n) equivariant graph neural networks. In International conference on machine learning (pp. 9323-9332). PMLR.

---

## Author Comment (AC2)

**Reviewer 2**

This is a review of the pre-print by Koo et al. in The Cryosphere titled "Calibrating calving parameterizations using graph neural network emulators: Application to Helheim Glacier, East Greenland". This study describes the use of Graph Neural Networks (GNNs) of various types to emulate the behavior of ISSM, a finite-element ice sheet model. There is also extensive comparison of the use of GNN to more traditional FCNs, which require input data to be on a uniform, rectangular grid. Once the NN is trained, it can be used to predict ice thickness, velocity and terminus position (through an ice mask) at a subsequent time step based on these fields at the prior time step and a parameter governing calving behavior in a model parameterization (sigma_max).

The most novel aspect of this study is the use of GNNs to emulate a finite-element ice sheet model. The study makes a good case for why this type of NN makes sense for emulating a model on a non-uniform mesh, though I'm not sure the way in which its accuracy was compated to FCNs is completely fair. In that sense, I found this the most compelling potential advance of this study to be the development of a general purpose ice-sheet-model emulator (like IGM, but with some advantages). I am less convinced that we have necessarily learned much about calving from the use of this new method. I explain some of my major issues in this regard below and then a list of smaller suggestions below that.

- Thank you for your constructive comments. We carefully reviewed all your comments and will handle them to improve our manuscript. Please see the below responses to your comments for details.

**Major points:**

1. As I read through the study, I found myself unclear about the scientific use of this methodological advance. The GNN will emulate ISSM at high accuracy and significantly lower computational cost. What questions will that help us to answer that isn't possible with conventional methods? This is a particularly important question to answer since this is submitted to The Cryosphere, a disciplinary journal, as opposed to a more methods oriented journal like GMD or JGR:MLC.

- The objective of this study is to facilitate the assimilation of numerical models and observations using GNN emulators. Since it is quite challenging to find optimal parameterizations of numerical models that are consistent with observations because of the computational demands of numerical models, we propose to use the GNN emulators to find appropriate parameterizations of numerical models. In the revised manuscript, we will highlight the objectives of this study and how this can contribute to scientific findings.

Once I got to the end, I saw that the main application this new emulator was used for was essentially something like transient parameter estimation (using the low cost of the NN to enable an exhaustive grid search for sigma_max at each time step). But then the result of this application didn't make physical sense. The calving front retreats while sigma_max increases, which is sort of opposite what should happen *if* calving drives retreat (which it may not). The text pushes off the explanation on "other processes" without much investigation of whether the methods may be at fault, or other potential explanations. Ultimately, this is a challenge of using completely data-driven ML without further investigation of the latent space of the NN - the emulator is a black box, so it is challenging to diagnose what is happening in it that causes this counter-intuitive result.

- It is true that the calving front retreats while sigma_max increases in this case, but we can also see the ice velocity increase during the same period. Mathematically speaking, as shown in Eqs. (1) and (2), the calving rate can increase with increasing ice velocity or increasing tensile stress. Meanwhile, decreasing sigma_max will increase the calving rate. In general, lower sigma_max means the ice front is easier or more susceptible to calve. However, as we showed in this manuscript, as the ice front retreats further upstream, sigma_max increases, which means the calving rate may decrease. We agree that the von Mises calving law is a simplification of reality, and this law certainly misses some important internal or external feedbacks. The result we showed in this study is the best parameterization to reproduce observations. We meant "other processes" to express the potential imperfection of the VM method. Although we chose the VM calving law for Helheim Glacier because the previous studies showed that this law fits the Helheim Glacier, this calving law would not be 100 % perfect to describe every detail of the calving process. In the revised manuscript, we will add discussions about the limitations of the VM calving law.

2. The study, as it stands, has not convinced me that the GNNs trained as they were in this study, generalize at all outside of the very limited training data. The test data is completely within the interior of the limited parameter/state space on which the GNNs are trained. If I simply used linear interpolation to generalize from the training data to the test data, how accurate would that be in comparison? It would certainly be computationally cheap.

- This is an interesting idea. In general, though, calving fronts do not always respond linearly to changes in sigma_max, and we find tipping points or thresholds. We agree that this is a good idea though and will add some experiments to the Appendix by applying a linear interpolation with neighboring σmax values as suggested, to see if the GNN is more accurate than a simple interpolation.

More importantly, the GNNs have not been tested on any cases that are out of the temporal or spatial sample of the training data. If the aim is to narrowly train the model to do a really good job learning what Helheim did from 2007 to 2020, thats OK, but state that narrow expectation

explicitly. There are places in the study where you say that these GNNs could be used to replace an ice sheet model more generally, or in future simulations, but you haven't really shown the ability of the GNNs to do that, since they haven't been tested outside of this very narrow place and time period.

- At this stage, we are just looking at a proof of concept that GNNs are viable and useful tools for parameter search. We only focus on how our GNN emulators work, specifically for Helheim Glacier, in the time periods from 2007 to 2020. In this study, we highlight the potential of EGCN "network architecture" to replicate the finite-element numerical ice sheet modeling, especially in delineating the calving front in the dynamic ice sheet system. Since this GNN or EGCN have not been used explicitly in ice sheet modeling, our study introduces a new useful tool to the field. However, we agree that the generalizability and time-invariant of the emulator are extremely important for further general applicability. This would be our next step based on our current results, and we will modify the network architecture to guarantee the generalizability of our emulator.

3. The accuracy metrics and differences therein are not very convincing. Interpreting a difference between 0.997 R value and 0.999 is not good statistics, particularly without assessing significance of these statistics on the training data. Similarly, I'm not sure how different a calving front accuracy of 98.6% vs. 99.4% is. I'm guessing both are significant at some very high level and so reading much into the difference beyond that isn't very meaningful. What happens if you drop some of the training data? Does the accuracy degrade? This is a common way to determine whether the NN has learned anything about the underlying dynamics of the system vs. acting as a fancy interpolator of the training data.

- This is a good point that was also mentioned by the first reviewer and will be revised. We found that the R values are too inflated because we calculated this metric for the entire glacier domain. However, the significant differences are only near the ice front and ice stream, while the other regions with slow ice show insignificant differences. We will recalculate these metrics only near the ice front and ice stream.

Additionally, the way that you train and then assess the accuracy of the FCN does not provide a fair comparison to the GNNs. By interpolating from the finite-element mesh to a uniform rectangular mesh, you've done two things: lowered the resolution of the training data in the finest parts of the grid and inflated the relative weight of the coarse parts of the grid by increasing the number of grid points in these areas. The places with the finest resolution in ISSM are also places where velocity is the highest and where the ice mask is changing (i.e. near the terminus) which will tend to make errors more important. effectively, after interpolating you have given the FCN worse training data than the GNNs. The least you can do is interpolate the FCN training data onto a uniform grid with resolution equal to the finest resolution in the ISSM mesh. Additionally, using some knowledge about where errors are likely to be the largest, you can apply weights in the FCN training loss function which are proportional to the finite-element grid

resolution. In that way, you will be "fixing" the mis-weighting that has occured by interpolating the training data that you then assess accuracy on.

I get that in some sense your whole point is that FCNs are not natural fits for finite-element training data, but with the relatively minor differences in accuracy you find, its hard to discern whether this is due to the NN being superior at capturing the data vs. the training data just being different due to interpolation artifacts. These are very different claims.

None of this changes the fact that GNNs are likely to be much more efficient at natively training and then running on the finite element mesh. I believe your case that they are computationally superior, but I'm not sure I see much difference (or a fair comparison of differences) in the accuracy. My suggestion is simply to focus on the fact that emulating finite-element models (which most modern ice sheet models are) is more natural using GNNs since it doesn't require interpolation and that the computational advantage of GNNs over FCNs is massive. The GNNs do a great job accurately emulating the model by any objective measure, so emphasize this.

- The need for additional interpolation from the finite-element mesh to a uniform rectangular mesh is one of the main reasons we claim that FCNs are not "natural" for emulating finite-element models, such as ISSM. As you point out, this additional interpolation brings significant problems in replicating finite-element models: (1) loss of detailed resolution in the finest element area with fast ice velocity (primarily near ice front and ice stream), (2) allocation of unnecessary computational loads to coarse-resolution areas where ice velocity is slow. We emphasize these limitations of FCN and the advantages of GNN in replicating "unstructured-mesh" simulations.
- Our current loss function, mean square error (MSE), already works as a sort of "weighting" loss function because this loss function weights the large error area by squaring the errors. Moreover, the modification of model weighting of FCN does not really solve the two problems of the FCN for finite-element model: (i) loss of details and (ii) inefficient allocation of computational resources. These problems are caused by the "fixed-resolution" nature of FCN for all locations, and increasing FCN resolution would bring an exponential increase in computational cost. However, in our preliminary research, we found that the GNN architecture is a "flexible-resolution" approach, which is not dependent on changing resolution. That is, even if we change the spatial resolution of unstructured meshes, the performance of GNN is not affected by mesh resolutions. The preprint version of our preliminary paper is available at this link: https://arxiv.org/abs/2402.05291
- In this aspect, we argue that the fact that we do not need to interpolate the model results on a regular grid is a significant advantage of GNNs.

**Minor suggestions:**

L1: Increasing calving has been linked to the retreat

- Yes, higher calving rates can lead to retreat. However, calving is also somehow responsible for the acceleration and thinning of glaciers, so we would like to keep them "separate" in the text as well.

L3: have been used to simulate ice

- Done.

L10: reproduce the observed evolution

- Done.

L22: total ice sheet mass loss

- Done.

L28,30: optimal in what sense?

- We meant the optimal parameterizations that match observations. We clarify it in L28.

L35: as a boundary condition in numerical

- Done.

L41: necessitate using high-performance

- Done.

L56: the training of emulators

- Done.

L60: outlet glaciers in Greenland

- Done.

L79: The migration rate of the ice front

- Done.

L82: ice front migration rate (velocity is confusing here because it could refer to other things)

- Done. Thank you for your suggestion.

L87: VM has not been defined as an acronym

- VM is defined in L29.

L87: correlates with weaker ice

- Done.

L89: many observational studies have found tensile strength as low as 100 kPa (Vaughn 1993 is a particularly well known paper), so I'm not sure where this lower bound is coming from.

- This lower bound value is from Morlighem et al. 2016 and Petrovic, 2003.

L91: is important to accurately reproducing observed glacier evolution

- Done.

L117: CNN cannot represent finite-element ice sheet models on their native grid

- Done.

L122: focused on calibrating calving parameterizations using

- Done.

L135: each transient simulation denerates of a total of 261 outputs between.

- Done.

L136: calibrated and held constant

- Done.

L136-140: the use of semicolons here is a bit challenging to read. Why not just write these as separate sentences?

- Done. We separate it into several sentences.

L146: adjacency matrices?

- Yes. We change it into adjacency matrices.

L151: we compare to remote-sensing

- Done.

L201: you aren't the first to develop an EGCN - you train a NN architecture that has previously been described in other papers

- We change "develop" to "adopt".

L242: don't you mean validation instead of testing on this line?

- Yes, corrected.

L243: Related to point #2 above - it seems that you have chosen test cases non-randomly, and I wonder what would happen if you chose 0.7 as a test case instead (with 0.7 not in the training/validation data)?

- This training-testing split checks the applicability of emulators for various σmax values to match the numerical simulation and observations. By applying the emulators trained with various σmax values to out-of-sample σmax values (0.75 and 0.90), the current approach and results show that our emulators can represent the ice sheet dynamics and calving at out-of-sample σmax values within the range of 0.70-1.10 MPa.
- In the revised manuscript, we will add some experiments in the Appendix to get insight into how reliable GNNs are in replicating the simulation results for two σmax: 0.75 MPa and 0.90 MPa. We will apply a linear interpolation with neighboring σmax values and compare this with GNNs: comparing σmax = 0.8 and 0.7 to the test set for σmax = 0.75, and comparing σmax = 0.85 and 0.95 to σmax = 0.90.

L271: remarkable in what sense? This is related to point #3 above - what is your benchmark that you are comparing to? Significance at 0.95 or above?

- We meant high R-values (significance at 0.99 for all cases) for replicating the spatial and temporal patterns of ice velocity and thickness. However, as we mentioned in point #3, these high R-values could be inflated because we calculated these metrics for the entire glacier domain. We will recalculate these metrics, covering only near the ice front and ice stream.

L305-315: it could be made clearer here that when tested on the exact same hardware, GNN are faster. Comparing wall time on two different processes or a different number of processors is not a fair comparison.

- Here, we should note that the ISSM runs on a high-performance computing (HPC) system because it is computationally demanding, while deep learning emulators (i.e., GNN and CNN) can be simply run on a local desktop. We would like to highlight that our deep-learning emulators have significant computational efficiency even on a local machine.

L350: I'm not sure I buy this argument partly because enough information hasn't been provided. Was ocean frontal melt included in the ISSM simulations? Do we know if melange increased at Helheim over these years? If so, it would presumably have an influence on calving rate, which could be captured effectively through sigma_max...This gets to the point above that this discussion here is entirely too brief and doesn't engage with any prior work on Helheim and its recent changes. If this paper is to be appropriate for TC, instead of say, GMD, then that discussion would be needed.

- Agreed. Since we only assume the VM calving law, we cannot really infer other factors besides the calving threshold. We will remove this part in the revised version. Instead, we will add some discussions about the limitations of the VM calving law, which we merely depend on to describe calving processes in this study.

L358: this begs the question: what would happen if you interpolated all the training data onto a rectangular grid, and then used that to train both the FCNs and the GNNs? This would be a fairer comparison than what you have now.

- Interpolating the training data onto a rectangular grid is not a good choice to delineate the calving front. It loses the detailed resolution at the calving front.

L364: CNN->FCN

- Done.

L373: why should GNNs be trained with numerical simulations?

- As "statistical emulators", GNNs should learn the statistical relationships between inputs and outputs represented in numerical simulations. We will add some explanation about this statement.

L385 with 13-year transient simulations of Helheim

- Done.

L391: how are these emulators promising for parameterizing future behavior? They provide no way of constraining sigma_max without observations and you haven't demonstrated that they can extrapolate outside the temporal sample of the training data. Perhaps they could be used to do uncertainty quantification since they enable cheap MCMC sampling of parameters space.

- Thank you for this point. Our emulator allows us to predict the next-time-step ice velocity and ice front conditions when the previous-time-step conditions are provided. Although we did not directly extrapolate the temporal sample of the training data out of 2007-2020, our emulator can predict the future behavior of ice sheets under certain conditions. As you suggested, we also agree that our cheap emulator enables uncertainty quantification via MCMC sampling.

**References**

- Choi, Y., Morlighem, M., Wood, M. & Bondzio, J. H. Comparison of four calving laws to model Greenland outlet glaciers. Cryosphere 12, 3735–3746, https://doi.org/10.5194/tc-12-3735-2018, 2018.

- Wilner, J. A., Morlighem, M. & Cheng, G. Evaluation of four calving laws for Antarctic ice shelves. Cryosphere 17, 4889–4901, 2023.
- Morlighem, M., J. Bondzio, H. Seroussi, E. Rignot, E. Larour, A. Humbert, and S. Rebuffi (2016), Modeling of Store Gletscher's calving dynamics, West Greenland, in response to ocean thermal forcing, Geophys. Res. Lett., 43, 2659–2666, doi:10.1002/2016GL067695.
- Petrovic, J.J. Review Mechanical properties of ice and snow. Journal of Materials Science 38, 1–6 (2003). https://doi.org/10.1023/A:1021134128038
- Velickovic, P., Cucurull, G., Casanova, A., Romero, A., Liò, P., and Bengio, Y.: Graph Attention Networks, https://doi.org/10.48550/arXiv.1710.10903, 2018
- Satorras, V.G., Hoogeboom, E. and Welling, M., 2021, July. E (n) equivariant graph neural networks. In International conference on machine learning (pp. 9323-9332). PMLR.

---

## Author Response (AR1)

We thank two reviewers for providing valuable and constructive comments to improve our manuscript. We made significant revisions to the manuscript based on the reviewers' comments. Please see our detailed responses to the reviewers as follows. We note that all specified line numbers are based on the **track-change version**.

**Reviewer 1**

In this manuscript, Koo et al. describe the application of several variants of a graph neural network to emulate ISSM at Helheim Glacier. I am unsure as to exactly what this emulator does- whether it is similar to IGM in producing an approximate solution operator to the (in this case coupled) momentum and mass conservation equations or whether it is more similar to He (2023) in being geometry specific- but in general terms it is trained to reproduce the predicted ice velocity and thickness as a function of time and space.

I think that this paper has the potential to be a useful contribution to the literature, and the goal of coming up with emulators that operate naturally in the same discrete setting as FEM-based ice sheet models is worthy. Additionally, the saliency with which the neural networks either learn or memorize (I'm not entirely sure which) the model's behavior is impressive. However, the manuscript needs significant clarification (and in some cases moderation) of its claims in order to assess their veracity and utility. In particular, the methods are unclear and not reproducible- as mentioned above, I am not clear what the features used for prediction actually are. Additionally, the paper does not include a fair representation of the computational costs of the proposed methods. Finally, the paper makes many claims that its proposed methodology is better than others without providing sufficient evidence to back up that claim. Detailed comments are below.

- Thank you for your constructive comments. We reviewed them carefully and have made necessary changes accordingly.

L63 Downs (2023), which is already referenced in this paper, provides significant insight into this question, and also serves as another example of a surrogate model being used to infer calving dynamics at Helheim Glacier (though not a GNN).

- This is a good point. We added brief explanations about how Downs et al. 2023 inferred the calving dynamics of Helheim Glacier, and how our approach differs (L46-58)

"Downs et al. (2023) used a Gaussian Process (GP) emulator to infer the sensitivity of time-independent model parameters to the frontal ablation of Helheim Glacier in Southeast Greenland. Although they were able to identify the best set of calving threshold parameters in the VM calving law, their artificial neural network (ANN) approach did not account for spatial relationships or the interactions between neighboring nodes.

> Additionally, their emulator focused on matching the observed and modeled terminus positions along a central flowline rather than the entire glacier system."

> "Given that ice sheet dynamics and iceberg calving are affected by the underlying bed topography, it is important for the emulators to learn the spatial context across the entire glacier domain. To account for spatial relationships between nodes for emulating ice sheet dynamics, prior research predominantly relied on Convolutional Neural Networks (CNNs) (Jouvet et al., 2022; Jouvet and Cordonnier, 2023). While promising, CNNs are tailored for regular Euclidean grid structures, such as images. This approach may therefore not be the optimal choice for capturing unstructured meshes that are typical of finite-element-based numerical models."

L75 The acronym VM should be defined here.

- VM is defined in L31.

L78 Physically, why should the stress threshold have to be calibrated on a glacier-by-glacier basis?

- Thank you for raising this important point. While the glacier-specific calibration of stress thresholds is beyond the central scope of this paper, we acknowledge the complexity behind this issue. Previous studies (e.g., Morlighem et al. 2019) found that most glaciers in Northwest Greenland had a threshold of around 1 MPa (see the first paragraph of their results section). However, they also found that some glaciers "needed" a different threshold, even though it is not fully clear why that is the case. Some of these variations could be due to a bias in the model initialization or the forcing, but also due to different levels of damage of individual glaciers, which may not be fully captured by such a simple parameterization. It is also not clear whether VM is always adequate to capture all the different modes of calving. We add a short explanation to the text (L81-82, L449-451).

Sec. 2.3 This section should include some earlier literature. It would be worth looking at Tarasov et al. (2012, https://doi.org/10.1016/j.epsl.2011.09.010) and Brinkerhoff et al. (2021, https://doi.org/10.1017/jog.2020.112).

- We added Tarasov et al. 2012 and Brinkerhoff et al. 2021 to Section 2.3.

L123 I don't understand why GNN's would be particularly suited for ice front migration relative to other architectures.

- We apologize for missing a detailed explanation. We meant that this was particularly attractive for finite-element ice sheet models because of how they typically discretize the model domain. ISSM is a numerical ice sheet model that relies on unstructured meshes: it uses a finer resolution in the fast-flowing region and a coarser resolution in slow-moving regions to optimize computational efficiency. However, CNNs inherently rely on a fixed

resolution (regular grid). In order to capture the calving dynamics at the ice front, we need a sufficiently fine resolution close to the terminus which will require the grid size of the CNN to be very fine over the whole domain. On the contrary, since GNNs directly use the ISSM unstructured meshes, they can keep the advantages of the ISSM numerical simulations in a more natural way. Hence, GNNs can be a better option to (1) obtain accurate ice front mitigation by embedding the interaction between neighboring nodes, (2) obtain a sufficiently fine resolution to capture ice front migration, (3) minimize the computational load. We add a detailed explanation of why we chose GNN as the backbone architecture. (e.g., L48-60, L70-75, L130-136)

L146 'adjacent' → 'adjacency'.

-   Done. (L188)

L149 Finite elements are, at their core, interpolants. Does bilinear interpolation here just mean that the FEM solution is evaluated at grid points? Or is some other interpolant introduced?

-   Yes. Since the FEM solution is not provided on a regular grid, we interpolate the velocity and thickness solutions on regular grid points.

L179 the square root does not need to be defined.

-   Done. (L222)

L180 'of' → 'with'.

-   Done. (L224)

L185-187 I don't think that this description makes sense. The architecture *for some particular hidden layer* weights different adjacent nodes differently for an operation that is otherwise the same as vanilla graph convolution. The resulting convolutions are then stacked– graph convolution is stacked, but the attention operation is internal to that.

-   Correct. This self-attention operation is internal to a hidden layer. We add some clarification about it. (L229)

Eq. 6: I am not sure whether this equation is correct, but I am also not sure whether it's necessary- it seems like maybe something very detailed that appears in the reference (although it does seem weird to concatenate the projected feature vectors like this, which would seem to make the attention scores sensitive to the order of arguments). Maybe okay to forego?

-   We used the same equation from the reference paper (Velickovic, 2018), but we removed this equation in the revised manuscript because it is just a detailed version of Eq. 5.

Fig. 3 I don't think that this figure is helpful for illustrating how either of these models work.

- This figure was intended to illustrate the concept of attention and equivariance. We removed this figure from the revised manuscript.

Sec. 4.3 This section is really difficult to understand. One thing that I can glean from this is that this is operation is O(n2) in the number of graph nodes- does that have any implications for performance?

- We have revised Section 4.3 to add some clarification about the EGCN model. We do not delve into every detail of how the EGCN works, but the details about the EGCN, including how this architecture guarantees equivariance, are well-documented in Satorras et al., 2021. For a more comprehensive explanation of how this architecture operates and its theoretical foundations, we refer readers to that source. (L274)
- EGCN is an operation of $O(n^2)$ because it operates on all graph nodes, while the GCN and GAT operate only on the adjacent nodes. That is, the EGCN operates on all graph nodes to preserve equivariance in the entire graph. Thus, the EGCN requires more processing time than the GCN and GAT, as shown in Section 5.2. We add more discussion about this in Section 5.2. (L367-369)

Sec. 4.5 This description of the model's inputs and outputs should appear at the beginning of the methods section. Furthermore, specifically what these features mean needs to be much more clearly described- as it stands, I cannot assess the quality of this work because, despite looking at both the manuscript and the linked code, I cannot tell what this emulator is building a mapping between. There are a few things implications to mention associated with this. First, it appears that the time t is explicitly included as a feature. This then implies that the surrogate is not time-invariant and the mapping should be thought of as a tool for downstream analysis (like He (2023) or Downs(2023)) rather than as learning the solution operator for Stokes' equations (or Stokes plus continuity since the present work claims to predict thickness as well). This is fine- such models can be very useful- but it mandates a change in language to reflect the fact that it is unlikely that this method can generalize to other locations or times. Second, the velocity components at some previous time (it's not clear whether this is from the model's previous time step or at the beginning of the simulation- this notation needs to be modified to be more clear) is used as a feature. This is an unfortunate choice because one thing that we know about ice physics is that the velocity is approximately diagnostic of the geometry- if you know the latter, you can predict the former. In the absence of that property, how can this model be started? Is it the case that ISSM has to be run first before this emulator can be applied? What is a 'forwarding process'? What is a graph 'structure'? Is this just the collection of node/edge features? This section is essential to understanding what is going on (more essential even than architecture choice), yet it's only two paragraphs long and does not have sufficient information to allow for reproducibility.

- Thank you for your detailed comment. Regarding the mapping of input and output features, Fig. 2 helps to understand how this emulator works. We add some explanations about the mapping via our GNN emulators in L283-292.
- Regarding your first point about the time-invariant, we agree that our emulator is not completely time-invariant. This study aims to find the parameterization that best matches real observations into numerical models. Therefore, we design our emulator to predict the "next-time-step" ice dynamics (at time *t*) from the "previous-time-step" velocity and ice geometry (at time *t-1*) with certain parameterization (specifically the calving threshold; sigma_max). This facilitates the comparison between emulator outputs and real observations and finding good parameterizations based on the statistical mapping manifested in the GNN emulators.
- Our emulators output four variables, the representative indicators of ice sheet dynamics: x-velocity, y-velocity, ice thickness, and ice front (Fig. 2). Although the x-velocity, y-velocity, and ice front observations were available regularly at every time step via satellite remote sensing, the regular observations of ice thickness across the entire glacier domain was not available. Thus, we decided to include time *t* as an input feature to indicate the temporal evolution of ice thickness instead of using ice thickness at *t-1* as an input variable. In this way, our emulator can indirectly see the impacts of the temporal evolution of ice thickness at a specific time step, even though we cannot directly use ice thickness observations. We add this explanation about input feature selection to Section 4.5 and mention that our method can have limitations when generalized to other locations or times (L283-292, L438-444).
- Here, the forwarding process indicates the determination of output from input features via a neural network. 'Graph structure' is the collection of nodes and edges. Please see L283-292 for revised explanations.

Eqs. 11, 12, 13 These are all standard definitions that do not need to be included here.

- While we recognize that Eqs. 11-13 are standard definitions, it is important to retain them for the sake of completeness. Including these equations ensures clarity, particularly for readers who may not be familiar with the specific metrics or may wonder about the exact calculation methods used in this study. This helps to maintain transparency and accessibility in our methodology.

L269 'out of sample' is perhaps a bit of an overstatement- the degree of correlation between neighboring $\sigma$ values would very likely be quite high- as a check, it would be interesting to see what error is induced by comparing model predictions made using $\sigma max = 0.8$ to the withheld test set values for $\sigma max = 0.75$ (or something like that). I expect the metrics would be similar because there is little difference between such small variations in the parameter. A more useful test would be to train on just the two extremal values (0.7,1.1) and see if it can still interpolate well.

- We agree that the degree of correlation between neighboring σ values can be high. In the Appendix, we add some additional experiments with neighboring σ values: comparing σmax = 0.8 and 0.7 to the withheld test set values for σmax = 0.75, and comparing σmax = 0.85 and 0.95 to σmax = 0.90.

Table 1 These metrics are all so inflated as to be useless. Is it possible to come up with relative metrics that use more significant digits? I also think it would be better to combine Tables 2 and 3 with Table 1- it is useful to think of model accuracy relative to model size and expense.

- We decided to use R because this metric provides the "statistical correlation" between predicted values and true values. That is, R provides insight into how the spatial or temporal patterns of prediction agree with those of ground truth, such as where/when the value is relatively high or low. Hence, if the spatial and temporal patterns are similar, a very high R-value is expected. The high R-values indicate that the deep-learning emulators replicate the spatial and temporal patterns of ice velocity and thickness. (L326)
- To avoid R values being too inflated, we recalculate these metrics only near the ice front and ice stream (Table 1). It is expected that significant differences between numerical simulations and emulators occur near the ice front and ice stream, while the other regions with slow ice show insignificant differences. Indeed, the R values of the FCN decrease significantly because the spatial resolution of the FCN is not enough to represent the high resolution near this fast ice region. Nevertheless, the R values of GNN emulators are still high (while RMSE values slightly increase), which indicates that GNN emulators can replicate the spatial and temporal patterns near the fast ice region as well.
- Thanks for the suggestion about tables. We combined Tables 2 and 3. However, combining Table 1 with the other two makes the table too distracting, as there is too much information.

Sec. 5.1 A single study does not establish the superiority of GNNs over CNNs for tasks such as these- it could (and very likely is) the case that the current results are incidental (or cherry picked) and that different researchers could find different conclusions. Furthermore, with respect to efficiency, there are many tricks that can be performed on CNNs to make them faster that have no analogue for an unstructured mesh, none of which were presumably included in the present analysis. The style of NeurIPS or similar notwithstanding, it is generally unhelpful to try to establish the primacy of one method over another in this way, and I would encourage the authors to either undertake a much more controlled and systematic comparison between methods or to reframe this as less of a competition.

- Thank you for pointing out this. In this paper, we would like to highlight that GNNs can be an alternative option for directly emulating the unstructured meshes of ISSM. The unstructured meshes of ISSM are characterized by variable resolutions, which allocate high resolution to fast ice region and low resolution to slow ice region. While GNNs can directly use the finite-element mesh data structures, CNNs cannot take direct advantage

of these unstructured mesh structures in resolution and computational efficiency as they should operate only on regular grids; CNNs also require further interpolation of unstructured meshes to regular grids. In other words, using CNNs as emulators for ISSM can introduce two problems: (1) the CNN grid with fixed resolution can lose dynamical details in fast ice areas; (2) the CNN grid requires unnecessary computational demands in slow ice areas.

- In the revised manuscript, we highlight the advantages of GNN, particularly for the ISSM ice sheet model with unstructured meshes. (L52-60, 70-75)
- We agree that several tricks can be applied to CNNs. However, in this study, we would like to approach this issue in terms of "network architecture" rather than tuning the CNN architectures. We add some clarification to the objectives of this study in L70-86.

Sec. 5.2 It is frankly absurd to not include even a mention of the computational cost associated with increasing the training data, which- so far as I can tell- must be repeated for any new geometry or parameter or location. Ignoring this cost does of course lead to much more impressive speed-ups, but these are not real. I would expect this problem to become considerably more severe when trying to use this technique to emulate models that are a function of more than a single parameter- the curse of dimensionality still applies. Furthermore, the notion that a GNN will be more suitable than a CNN for higher resolution modelling is another strawman because it ignores the fact that generating the cost of generating the training data (which is presumably more expensive than any network training) is also going to scale proportionally with resolution. I would encourage the authors to revisit this entire section with a more sober perspective aimed at delivering a factual assessment of the present work's utility.

- We agree that the repeated application of our GNN emulators to another location or parameter would require more computational cost. In this study, our simulation is only focusing on the VM calving law that requires only $\sigma_{max}$ for parameterization; however, if we use another calving law that requires additional parameters, it would take much more time to collect the simulation data from various parameter settings. Such computational demands with multiple parameters and different locations can be inferred from our results. Additionally, we want to emphasize that the computational significance of deep learning emulators is that they can find the statistical relationships between input and output features and the optimal parameter settings once they are trained from the provided simulation data. We add a brief mention of this problem in Section 5.2 L382-384.
- Generating training data is not a big task for GNNs. Since GNNs directly use the mesh of ISSM, there is no significantly time-consuming extra workload. The only thing we need to do to generate a training dataset for GNN is to define the graph structures with the ISSM meshes, elements, and adjacency matrix of nodes, and it takes only a few seconds in a local machine. On the contrary, this additional workload is applied to CNNs rather

than GNNs because CNNs require interpolating all the features from the unstructured mesh of ISSM into regular grids.

L351 This is only true if the model's behavior in response to variations in this new parameter is as well quantified as it is to the parameter considered in this work, which may not be true, or may not be tractable.

- Agreed. Since it is not clear how the other parameters, which are not considered in our emulator, impact calving, we remove this part in the revised version. Instead, we briefly mention the necessity of evaluation of GNN architectures for the other calving processes or glaciers (L382-384).

Sec. 6 Again, I strongly urge the authors not to try to cast their work in terms of 'superiority'- the present work does not provide sufficient evidence for such blanket statements, nor is it necessary.

- We agree that this was a poor choice of word, and perhaps an unnecessary comparison. We tone down this point in the manuscript as suggested (L415-416).

Fig. 7 This needs a more descriptive caption- I am struggling to see what these figures are showing.

- Sorry for the too-short caption. We add a more descriptive caption for this figure (Figure 6).

L370–371 Is this statement really necessary?

- We remove this statement in the revised version.

**References**

- Choi, Y., Morlighem, M., Wood, M. & Bondzio, J. H. Comparison of four calving laws to model Greenland outlet glaciers. Cryosphere 12, 3735–3746, https://doi.org/10.5194/tc-12-3735-2018, 2018.
- Wilner, J. A., Morlighem, M. & Cheng, G. Evaluation of four calving laws for Antarctic ice shelves. Cryosphere 17, 4889–4901, 2023.
- Morlighem, M., J. Bondzio, H. Seroussi, E. Rignot, E. Larour, A. Humbert, and S. Rebuffi (2016), Modeling of Store Gletscher's calving dynamics, West Greenland, in response to ocean thermal forcing, Geophys. Res. Lett., 43, 2659–2666, doi:10.1002/2016GL067695.
- Petrovic, J.J. Review Mechanical properties of ice and snow. Journal of Materials Science 38, 1–6 (2003). https://doi.org/10.1023/A:1021134128038
- Velickovic, P., Cucurull, G., Casanova, A., Romero, A., Liò, P., and Bengio, Y.: Graph Attention Networks, https://doi.org/10.48550/arXiv.1710.10903, 2018

- Satorras, V.G., Hoogeboom, E. and Welling, M., 2021, July. E (n) equivariant graph neural networks. In International conference on machine learning (pp. 9323-9332). PMLR.

**Reviewer 2**

This is a review of the pre-print by Koo et al. in The Cryosphere titled "Calibrating calving parameterizations using graph neural network emulators: Application to Helheim Glacier, East Greenland". This study describes the use of Graph Neural Networks (GNNs) of various types to emulate the behavior of ISSM, a finite-element ice sheet model. There is also extensive comparison of the use of GNN to more traditional FCNs, which require input data to be on a uniform, rectangular grid. Once the NN is trained, it can be used to predict ice thickness, velocity and terminus position (through an ice mask) at a subsequent time step based on these fields at the prior time step and a parameter governing calving behavior in a model parameterization (sigma_max).

The most novel aspect of this study is the use of GNNs to emulate a finite-element ice sheet model. The study makes a good case for why this type of NN makes sense for emulating a model on a non-uniform mesh, though I'm not sure the way in which its accuracy was compared to FCNs is completely fair. In that sense, I found this the most compelling potential advance of this study to be the development of a general purpose ice-sheet-model emulator (like IGM, but with some advantages). I am less convinced that we have necessarily learned much about calving from the use of this new method. I explain some of my major issues in this regard below and then a list of smaller suggestions below that.

- Thank you for your constructive comments. We carefully reviewed all your comments and improved our manuscript accordingly.

**Major points:**

1. As I read through the study, I found myself unclear about the scientific use of this methodological advance. The GNN will emulate ISSM at high accuracy and significantly lower computational cost. What questions will that help us to answer that isn't possible with conventional methods? This is a particularly important question to answer since this is submitted to The Cryosphere, a disciplinary journal, as opposed to a more methods oriented journal like GMD or JGR:MLC.

- The objective of this study is to facilitate the assimilation of observations in numerical models using GNN emulators. Since it is quite challenging to find optimal parameterizations of numerical models that are consistent with observations because of the computational demands of numerical models, we propose to use GNN emulators to find appropriate parameterizations in numerical models. In the revised manuscript, we highlight the objectives of this study and how this approach can contribute to scientific findings.

  "As the main parameter that determines the terminus positions, the temporal variations of calving parameters and their impacts on calving should be explored (Downs et al., 2023).

In this study, we train GNN models using simulation data derived from a numerical model and evaluate their fidelity and computational efficiency in modeling the dynamics and calving front migration of Helheim Glacier. We assess the potential of GNN architectures as statistical emulators for numerical finite-element ice sheet models to represent spatial features of ice sheet dynamics and calving across the entire glacier domain." (L81-85)

Once I got to the end, I saw that the main application this new emulator was used for was essentially something like transient parameter estimation (using the low cost of the NN to enable an exhaustive grid search for sigma_max at each time step). But then the result of this application didn't make physical sense. The calving front retreats while sigma_max increases, which is sort of opposite what should happen *if* calving drives retreat (which it may not). The text pushes off the explanation on "other processes" without much investigation of whether the methods may be at fault, or other potential explanations. Ultimately, this is a challenge of using completely data-driven ML without further investigation of the latent space of the NN - the emulator is a black box, so it is challenging to diagnose what is happening in it that causes this counter-intuitive result.

- It is true that the calving front retreats while sigma_max increases in this case, but we can also see the ice velocity increase during the same period. As shown in Fig. 3, under a low calving threshold, a higher calving rate results in the acceleration of ice velocity, which can accelerate more calving again (Eqs. 1 and 2). That is, according to the VM calving law, the calving rate is dependent on both (i) calving threshold and (ii) ice velocity, and the feedback between them determines the ice terminus positions. Therefore, fine-tuning the calving parameter is the process of balancing the calving threshold with ice velocity that agrees with the observations. With a given increasing ice velocity, our fine-tuning process just finds the best threshold that aligns with the observations. If we set this threshold value too low, the ice front should be further retreated compared to the observation. Although the VM calving law is a simplification of reality and misses some important internal or external feedback, the results of this study just find the best parameterization to reproduce observations.
- We meant "other processes" to express the potential imperfection of the VM method. Although we chose the VM calving law for Helheim Glacier because the previous studies showed that this law fits the Helheim Glacier, this calving law would not be 100 % perfect to describe every detail of the calving process. In the revised manuscript, we add discussions about the limitations of the VM calving law in L449-451.

2. The study, as it stands, has not convinced me that the GNNs trained as they were in this study, generalize at all outside of the very limited training data. The test data is completely within the interior of the limited parameter/state space on which the GNNs are trained. If I simply used linear interpolation to generalize from the training data to the test data, how accurate would that be in comparison? It would certainly be computationally cheap.

- This is an interesting idea. In general, though, calving fronts do not respond linearly to changes in sigma_max. We add some experiment results to Appendix A by applying a linear interpolation with neighboring σmax values as suggested and check if the GNNs are more accurate than a simple interpolation.

More importantly, the GNNs have not been tested on any cases that are out of the temporal or spatial sample of the training data. If the aim is to narrowly train the model to do a really good job learning what Helheim did from 2007 to 2020, thats OK, but state that narrow expectation explicitly. There are places in the study where you say that these GNNs could be used to replace an ice sheet model more generally, or in future simulations, but you haven't really shown the ability of the GNNs to do that, since they haven't been tested outside of this very narrow place and time period.

- At this stage, we are just looking at a proof of concept that GNNs are viable and useful tools for parameter search. We only focus on how our GNN emulators work, specifically for Helheim Glacier, for the time periods 2007-2020. In this study, we highlight the potential of EGCN "network architecture" to replicate the finite-element numerical ice sheet modeling, especially in delineating the calving front in the dynamic ice sheet system. Since this GNN or EGCN has not been used explicitly in ice sheet modeling, our study introduces a new useful tool to the field. The generalizability and time-invariant of the emulator are extremely important for further general applicability. This would be our next step based on our current results, and we modify the network architecture to guarantee the generalizability of our emulator.

3. The accuracy metrics and differences therein are not very convincing. Interpreting a difference between 0.997 R value and 0.999 is not good statistics, particularly without assessing significance of these statistics on the training data. Similarly, I'm not sure how different a calving front accuracy of 98.6% vs. 99.4% is. I'm guessing both are significant at some very high level and so reading much into the difference beyond that isn't very meaningful. What happens if you drop some of the training data? Does the accuracy degrade? This is a common way to determine whether the NN has learned anything about the underlying dynamics of the system vs. acting as a fancy interpolator of the training data.

- R value provides the "statistical correlation" between predicted values and true values. That is, R provides insight into how the spatial or temporal patterns of prediction agree with those of ground truth, such as where/when the value is relatively high or low. Hence, if the spatial and temporal patterns are similar, a very high R-value is expected. The high R-values ~ 0.99 indicate that the deep-learning emulators replicate the spatial and temporal patterns of ice velocity and thickness. In terms of this spatial and temporal pattern, all GNN emulators do not have significant differences. However, we can evaluate the accuracy of "absolute values" using RMSE values. We highlight that the performance

comparison between different models should be based on RMSE rather than R (L313-315).

- R and BiAcc values could be too inflated because we calculated this metric for the entire glacier domain in the previous version. We recalculate these metrics only near the ice front and ice stream, where significant differences between numerical simulations and emulators mainly occur (Table 1). Indeed, the R values of the FCN decrease significantly because the spatial resolution of the FCN is not enough to represent the high resolution near this fast ice region. Nevertheless, the R values of GNN emulators are still high (while RMSE values slightly increase), which indicates that GNN emulators can replicate the spatial and temporal patterns near the fast ice region as well.

Additionally, the way that you train and then assess the accuracy of the FCN does not provide a fair comparison to the GNNs. By interpolating from the finite-element mesh to a uniform rectangular mesh, you've done two things: lowered the resolution of the training data in the finest parts of the grid and inflated the relative weight of the coarse parts of the grid by increasing the number of grid points in these areas. The places with the finest resolution in ISSM are also places where velocity is the highest and where the ice mask is changing (i.e. near the terminus) which will tend to make errors more important. effectively, after interpolating you have given the FCN worse training data than the GNNs. The least you can do is interpolate the FCN training data onto a uniform grid with resolution equal to the finest resolution in the ISSM mesh. Additionally, using some knowledge about where errors are likely to be the largest, you can apply weights in the FCN training loss function which are proportional to the finite-element grid resolution. In that way, you will be "fixing" the mis-weighting that has occurred by interpolating the training data that you then assess accuracy on.

I get that in some sense your whole point is that FCNs are not natural fits for finite-element training data, but with the relatively minor differences in accuracy you find, its hard to discern whether this is due to the NN being superior at capturing the data vs. the training data just being different due to interpolation artifacts. These are very different claims.

None of this changes the fact that GNNs are likely to be much more efficient at natively training and then running on the finite element mesh. I believe your case that they are computationally superior, but I'm not sure I see much difference (or a fair comparison of differences) in the accuracy. My suggestion is simply to focus on the fact that emulating finite-element models (which most modern ice sheet models are) is more natural using GNNs since it doesn't require interpolation and that the computational advantage of GNNs over FCNs is massive. The GNNs do a great job accurately emulating the model by any objective measure, so emphasize this.

- The need for additional interpolation from the finite-element mesh to a uniform rectangular mesh is one of the main reasons we claim that FCNs are not "natural" for emulating finite-element models, such as ISSM. As you point out, this additional interpolation brings significant problems in replicating finite-element models: (1) loss of

detailed resolution in the finest element area with fast ice velocity (primarily near ice front and ice stream), (2) allocation of unnecessary computational loads to coarse-resolution areas where ice velocity is slow. We emphasize these limitations of FCN and the advantages of GNN in replicating "unstructured-mesh" simulations. (see L70-75)

- Our current loss function, mean square error (MSE), already works as a sort of "weighting" loss function because this loss function weights the large error area by squaring the errors. Moreover, the modification of model weighting of FCN does not really solve the two problems of the FCN for finite-element model: (i) loss of details and (ii) inefficient allocation of computational resources. These problems are caused by the "fixed-resolution" nature of FCN for all locations, and increasing FCN resolution would bring an exponential increase in computational cost.
- In this aspect, we argue that the fact that we do not need to interpolate the model results on a regular grid is a significant advantage of GNNs.

**Minor suggestions:**

L1: Increasing calving has been linked to the retreat

- Yes, higher calving rates can lead to retreat. However, calving is also somehow responsible for the acceleration and thinning of glaciers, so we would like to keep them "separate" in the text as well.

L3: have been used to simulate ice

- Done. (L3)

L10: reproduce the observed evolution

- Done. (L11)

L22: total ice sheet mass loss

- Done. (L23)

L28,30: optimal in what sense?

- We meant the optimal parameterizations that match observations. We clarify it in L29-30.

L35: as a boundary condition in numerical

- Done. (L38)

L41: necessitate using high-performance

- Done. (L43)

L56: the training of emulators

- Done. (L67)

L60: outlet glaciers in Greenland

- Done. (L76)

L79: The migration rate of the ice front

- Done. (L100)

L82: ice front migration rate (velocity is confusing here because it could refer to other things)

- Done. Thank you for your suggestion. (L103)

L87: VM has not been defined as an acronym

- VM is defined in L31.

L87: correlates with weaker ice

- Done. (L108-109)

L89: many observational studies have found tensile strength as low as 100 kPa (Vaughn 1993 is a particularly well known paper), so I'm not sure where this lower bound is coming from.

- This lower bound value is from Morlighem et al. 2016 and Petrovic, 2003.

L91: is important to accurately reproducing observed glacier evolution

- Done. (L112-113)

L117: CNN cannot represent finite-element ice sheet models on their native grid

- This part is replaced with "CNN cannot take full advantage of finite-element ice sheet models on their native grid". (L132-133)

L122: focused on calibrating calving parameterizations using

- This part is removed.

L135: each transient simulation denerates of a total of 261 outputs between.

- Done. (L177-178)

L136: calibrated and held constant

- Done. (L178)

L136-140: the use of semicolons here is a bit challenging to read. Why not just write these as separate sentences?

- Done. We separate it into several sentences. (L179-181)

L146: adjacency matrices?

- Yes. We change it into adjacency matrices. (L188)

L151: we compare to remote-sensing

- Done. (L193)

L201: you aren't the first to develop an EGCN - you train a NN architecture that has previously been described in other papers

- We change "develop" to "adopt". (L247)

L242: don't you mean validation instead of testing on this line?

- Yes, corrected. (L297)

L243: Related to point #2 above - it seems that you have chosen test cases non-randomly, and I wonder what would happen if you chose 0.7 as a test case instead (with 0.7 not in the training/validation data)?

- This training-testing split checks the applicability of emulators for various σmax values to match the numerical simulation and observations. By applying the emulators trained with various σmax values to out-of-sample σmax values (0.75 and 0.90), the current approach and results show that our emulators can represent the ice sheet dynamics and calving at out-of-sample σmax values within the range of 0.70-1.10 MPa.
- In the revised manuscript, we add some experiments in the Appendix to get insight into how reliable GNNs are in replicating the simulation results for two σmax: 0.75 MPa and 0.90 MPa. We apply a linear interpolation with neighboring σmax values and compare this with GNNs: comparing σmax = 0.8 and 0.7 to the test set for σmax = 0.75, and comparing σmax = 0.85 and 0.95 to σmax = 0.90.

L271: remarkable in what sense? This is related to point #3 above - what is your benchmark that you are comparing to? Significance at 0.95 or above?

- We meant high R-values (significance at 0.99 for all cases) for replicating the spatial and temporal patterns of ice velocity and thickness. (L326)

L305-315: it could be made clearer here that when tested on the exact same hardware, GNN are faster. Comparing wall time on two different processes or a different number of processors is not a fair comparison.

- Here, we should note that the ISSM runs on a high-performance computing (HPC) system because it is computationally demanding, while deep learning emulators (i.e., GNN and CNN) can be simply run on a local desktop. We would like to highlight that our deep-learning emulators have significant computational efficiency even on a local machine. (L371-375)

L350: I'm not sure I buy this argument partly because enough information hasn't been provided. Was ocean frontal melt included in the ISSM simulations? Do we know if melange increased at Helheim over these years? If so, it would presumably have an influence on calving rate, which could be captured effectively through sigma_max...This gets to the point above that this discussion here is entirely too brief and doesn't engage with any prior work on Helheim and its recent changes. If this paper is to be appropriate for TC, instead of say, GMD, then that discussion would be needed.

- Agreed. Since we only assume the VM calving law, we cannot really infer other factors besides the calving threshold. We remove this part in the revised version. Instead, we add some discussions about the limitations of the VM calving law, which we merely depend on to describe calving processes in this study. (L449-451)

L358: this begs the question: what would happen if you interpolated all the training data onto a rectangular grid, and then used that to train both the FCNs and the GNNs? This would be a fairer comparison than what you have now.

- Interpolating the training data onto a rectangular grid is not a good choice to delineate the calving front. It loses the detailed resolution at the calving front.

L364: CNN->FCN

- Done. (L425)

L373: why should GNNs be trained with numerical simulations?

- As "statistical emulators", GNNs should learn the statistical relationships between inputs and outputs represented in numerical simulations. We add some explanation about this statement on L290-292.

L385 with 13-year transient simulations of Helheim

- Done. (L455)

L391: how are these emulators promising for parameterizing future behavior? They provide no way of constraining sigma_max without observations and you haven't demonstrated that they can extrapolate outside the temporal sample of the training data. Perhaps they could be used to do uncertainty quantification since they enable cheap MCMC sampling of parameters space.

- Thank you for this point. Our emulator allows us to predict the next-time-step ice velocity and ice front conditions when the previous-time-step conditions are provided. Although we did not directly extrapolate the temporal sample of the training data out of 2007-2020, our emulator can predict the future behavior of ice sheets under certain conditions. As you suggested, we also agree that our cheap emulator enables uncertainty quantification via MCMC sampling.

**References**

- Choi, Y., Morlighem, M., Wood, M. & Bondzio, J. H. Comparison of four calving laws to model Greenland outlet glaciers. Cryosphere 12, 3735–3746, https://doi.org/10.5194/tc-12-3735-2018, 2018.
- Wilner, J. A., Morlighem, M. & Cheng, G. Evaluation of four calving laws for Antarctic ice shelves. Cryosphere 17, 4889–4901, 2023.
- Morlighem, M., J. Bondzio, H. Seroussi, E. Rignot, E. Larour, A. Humbert, and S. Rebuffi (2016), Modeling of Store Gletscher's calving dynamics, West Greenland, in response to ocean thermal forcing, Geophys. Res. Lett., 43, 2659–2666, doi:10.1002/2016GL067695.
- Petrovic, J.J. Review Mechanical properties of ice and snow. Journal of Materials Science 38, 1–6 (2003). https://doi.org/10.1023/A:1021134128038
- Velickovic, P., Cucurull, G., Casanova, A., Romero, A., Liò, P., and Bengio, Y.: Graph Attention Networks, https://doi.org/10.48550/arXiv.1710.10903, 2018
- Satorras, V.G., Hoogeboom, E. and Welling, M., 2021, July. E (n) equivariant graph neural networks. In International conference on machine learning (pp. 9323-9332). PMLR.

---

## Referee Report (RR2)

**Review of Koo et al. (2024) 'Calibrating calving parameterizations using graph neural network emulators: Application to Helheim Glacier, East Greenland'**

**Summary**

I reviewed this paper as part of a second-round review; I was not involved in the first round of review. The paper demonstrates the application of graph neural networks as emulators for ISSM unstructured-grid simulations of Helheim Glacier, showing that the GNNs are able to accurately emulate the model outputs. The authors then go on to show that the much quicker run time of the emulator allows them to easily determine the required sigma_max parameter in a von Mises calving law to reproduce observed calving-front positions at Helheim between 2007 and 2020. They also compare their results to those achieved using a convolutional neural network, of the type previously used in ice-flow modelling.

I am honestly unsure what to make of this paper. I think there is a good GMD article in there about the application of GNNs as emulators for numerical glaciological models that use an unstructured grid – the machine-learning part is well-executed and makes more sense following the clarifications added in response to the first-round reviews – but, glaciologically, the paper fails to prove anything much. The authors appear to have a hang-up on proving that their approach is better than using a CNN, which is self-evident, as using a CNN to emulate an unstructured-grid model would be a poor choice to start with, but the authors perform a bad-faith comparison between their approach and using a CNN to do just that, and then keep mentioning at every available opportunity how the CNN was much worse than their GNNs. Both the previous reviewers pointed this out and I'm pointing it out again: either do the comparison in a fair manner or delete it entirely. As it stands, it makes the authors seem absurdly competitive about something no one else was competing over. On the glaciological side, the main finding seems to be – I think unintentionally – that the von Mises calving law isn't very good as a physical basis for calving. This might also be considered a little obvious: no existing calving parameterisations do a good job. I am also unconvinced that the emulator would do well at predicting the calving-front position at another glacier, or at Helheim at a different time, further limiting its glaciological relevance. At least, the authors provide no information or examples showing that their method would yield good results in such a case. I again feel I'm not saying anything particularly new compared to the first-round reviewers, but it bears restating.

 My recommendation would be to take out the comparison to a CNN and the glaciological interpretation, which is limited, unconvincing, and feels like an afterthought, and submit the core paper about the technical advance of applying GNNs successfully to an ice-flow model for the first time to GMD. If the authors want this to be published in a disciplinary journal, there is a substantial amount of work that needs to be undertaken, and I think it would be a case of revise and resubmit, as it would be too much to do within a major revisions timeline. I would also like to record my disappointment that I'm having to essentially restate many of the points raised by the first-round reviewers, as the authors seem to have not properly engaged with the review process beyond clarifying their own method, which was an important point raised by the initial reviewers, but by no means the only one, nor the one that was most damaging to the paper. I've consequently left this review in a more aggrieved tone than I would usually adopt in the hope that it communicates to the authors that they cannot brush these issues off and that substantial work is needed to properly consider and address them.

Page and line numbers refer to those in the clean version of the submitted manuscript.

**Major Comments**

- Comparison to FCN: see my increasingly tetchy comments below, but, as it stands, the comparison to the FCN tells us nothing about the relative performance of it or the GNNs and, worse, gives the whole paper an overly competitive tone that reflects badly on the authors and detracts from the more sensible bits of the paper. The comparison either needs to be done in such a way that it isn't just proving interpolation reduces numerical accuracy, or it should be abandoned. I would argue for the latter because I don't think it's needed: it's clear that GNNs are well-suited to this application because they can run on an unstructured grid, and, for that very reason, one wouldn't try to apply a CNN of some kind.

- Glaciological interpretation: The current interpretation of the results is unconvincing and extremely superficial. It also contradicts the expected behaviour of sigma_max as outlined by the authors themselves in the methods section, but no real explanation or discussion of this is provided in the paper (it is in the response to Reviewer 2, but the paper hasn't been changed to reflect it). As I've gone into in more detail below, I'm fairly certain the underlying problem is that the von Mises law is fundamentally quite bad as a physical representation of calving, and that therefore attempting to find a physical explanation for changes in sigma_max is a wild goose chase. It may be that the authors can come up with a convincing physical explanation, but they certainly haven't yet, so they either need to put the work in to do so, or abandon the glaciological interpretation and send this off to GMD as a technical modelling paper.
- Generalisability of the emulator: the authors are quite cagey about this, but all the results as presented in this paper prove is that the GNNs do a good job of emulating the specific ISSM simulations used to train them (or, to be fairer, ISSM simulations at Helheim within the parameter space, or very close to the parameter space, defined by the training data). I get no sense of whether they would perform well if applied to Helheim at a different time, or to another location, which again limits the glaciological interest of the paper. They may well perform well, but the paper doesn't show this. Again, unless the authors are prepared to do some substantial additional work showing a second application and proving that the emulator still does a good job, this paper really should go to GMD as purely a technical advance in applying GNNs to glaciological simulations. As a related issue, if, as the authors themselves seem to admit in their response to Reviewer 2, and as the results of this paper seem to show, sigma_max is not in fact actually physically meaningful but just a numerical fudge factor to get the model to agree with observations, how can the emulator be generalisable as there is quite possibly no consistent underlying pattern to learn that could be extrapolated to another time or place (particularly to the future where there would be no observations)? If the purpose of the emulator is to find the best value of sigma_max, but it has to be retrained for each new glacier or period, then, glaciologically, what is the usefulness of the emulator if you've got to run the numerical simulations anyway? I agree GNNs seem to be promising as a way of emulating ice-sheet models – that is a nice technical advance – but this particular application of them does not seem overly useful as presented

**Minor Comments**
- p.6, l.125: Delete 'merely' because a) it doesn't really make sense here and b) it makes it seem as if the authors are saying models on regular grids are rubbish, which is perhaps not an ideal tone to strike
- p.12, l.284-286: Don't think this paragraph really needs to be here, unless the authors are taking an extremely dim view of the intelligence or memory of their readers!
- Table 1: Sorry to bang on about this, both of the first-round reviewers having raised this point, but is the R metric really showing us anything useful, beyond all three GNNs are better than the FCN (which is obvious from the RMSE and calving-front accuracy anyway)? The R numbers for the GNNs are all so similar as to be quantitatively meaningless and I'd be wary about placing too much emphasis on very slightly different values for the third decimal place.
- p.13, l.298-303: Yes, these are fair criticisms of the fixed grid used by a CNN, but as both the previous reviewers state, the FCN is being set up to fail because the training data is interpolated onto the fixed grid at the start, introducing errors, and the results are then interpolated back onto the unstructured ISSM grid for the comparison, introducing more errors. In that situation, the FCN is mathematically virtually guaranteed to do worse. I do not doubt that the GNNs are a more natural fit for an unstructured grid and perform better on it, but if the authors want to compare the performance of the GNNs to an FCN, the FCN needs to be trained with data produced on and results evaluated on a structured grid. Otherwise, this comparison boils down to 'interpolation is bad for numerical accuracy', which isn't the most striking finding in the world. To be honest, is there even any need to compare to a fixed-grid CNN? The rest of Table 1 shows that the GNNs are doing a good job on the unstructured grid, which is the important thing; trying to prove that they're somehow inherently better than a CNN seems unnecessary, especially when the application presented here is clearly not one a CNN would be used for, because ISSM runs on an unstructured grid. Even if the authors have some particular animus against CNNs (I can't help feeling there's maybe a little bit too much of an attempt to prove that the method here is inherently better than IGM's, which is an unhelpful attitude), I would strongly suggest scrapping this comparison entirely, and limiting it to the existing

earlier remarks about how GNNs are a natural fit for an unstructured grid and that a CNN would be inappropriate for this application because it would require a fixed grid

- p.15, l.324: Yes, but compared to what? Presumably, given what follows, actually solving the model on CPUs, but then this advantage is true of all neural networks, not specifically GNNs. Some rephrasing might be needed here to make it clear which advantages are generic to neural networks and which are specific to GNNs
- p.15, l.332: I'm assuming there isn't any interpolation time being counted in the FCN stat here? Otherwise, again, not a fair comparison.
- p.17, l.346: ...possibly, the FCN is taking longer because the data it's being fed are inherently worse-quality because they've been interpolated? Though I admit that the magnitude of the speed-up in training is such that I don't doubt it's real, but my point again is that the comparison is essentially meaningless as presented
- Figure 3 and Figure 4: They're a bit much. I might suggest reducing the information overload by just having a nice six-panel figure of ISSM field, best-performing GNN field, difference, for the two parameter values, showing that the GNNs are doing a good job, and then sticking the 72-panel monsters in the supplementary information so that readers don't just glaze over with eye strain in the main paper itself. Something on the scale of Figure 5 is a much nicer presentation!
- p.17, l.349-354: I really think the CNN-bashing is getting a bit silly here, especially given how flawed the comparative basis in the paper is. See my earlier comments, but unless the authors are going to put in the work to do an actual fair comparison, statements like this are built on sand and make the whole paper seem weirdly aggressive. Do the work or delete the comparison entirely. If, after a fair comparison, GNNs do just turn out to be better, bash away, but right now, this is an untenable claim. The most that can be said would be something like 'While we have not conducted a full-scale comparison between GNNs and CNNs, owing to the difficulties introduced by the fundamentally different grid requirements, our successful emulation of ISSM shows that GNNs are inherently well-suited to replicating the results of finite-element models that use an unstructured grid, an application where a CNN, with its requirement for a structured grid, would struggle.'
- p.18, l.371-376: This feels like a bit of an afterthought. And also doesn't hold water. A higher sigma_max value should mean stronger, more stable ice (calving is more difficult), as the authors state on p.5, l.101-2. Here, sigma_max is increasing after 2014 as the calving front retreats rapidly, which is a contradiction in terms. As a related point, the ISSM simulations include ocean thermal forcing, so the authors should be able to say confidently if that's important here too, and it should be easy from remote-sensing observations to work out if there was more mélange after 2014, rather than the current weak formulation. Regardless, either the authors have to come up with an explanation of why a rapidly retreating calving front is, against all expectations, actually one more resistant to calving, or admit that sigma_max is not really physically based and is just a tuning parameter that is compensating for other errors and processes not included in the model, in which case interpreting changes in it is worthless. The latter is essentially what the authors do in their response to the same point raised by Reviewer 2, but the text here (and the wider paper) should be changed to reflect that, rather than the current presentation of the issue, which attempts to present the parameter as physically meaningful despite the evidence of this paper's own results and the authors' response to the previous round of reviews
- p.18, l.378-388: I'll let the authors interpolate my comment from those above: this is a use case where obviously using a CNN is a silly idea, and the comparison isn't fair, so there's really no need to keep sniping at them constantly. At this point, an equivalent argument is 'we proved this screwdriver was better at driving screws than this dead fish.' I mean, great, but people might think it was a bit odd that the dead fish was considered a sensible comparison in the first place
- p.20, l.406: This is probably why the earlier interpretation of the changes in sigma_max doesn't make sense
- p.20, l.408: Do they? It's an emulator: by definition, it's not going to tell anyone anything much about the underlying processes or mechanisms. I certainly can't say that I feel I've found out anything about calving mechanisms so far.
- p.20, l.409-411: The emulator isn't really emulating calving processes, because this ISSM setup is not modelling calving processes. The emulator is emulating a parameterisation of those processes, a parameterisation that explicitly ignores all the processes going on in favour of having a single easy parameter to play about with. The VM calving law is not itself a calving process, it's merely a (flawed) representation of underlying processes that are being ignored. Please be more careful with

the language here and make it clear what the emulator has done and can do (emulate ISSM solutions and determine the correct parameter values to match observed calving front positions), and what it can't do (provide any information at all about any of the underlying processes, especially if sigma_max isn't really all that physical)

---

## Referee Report (RR3)

**Review of Koo et al. (2024) 'Calibrating calving parameterizations using graph neural network emulators: Application to Helheim Glacier, East Greenland'**

**Summary**

I reviewed this paper as part of a third-round of review; I had previously reviewed it in its second round. The paper demonstrates the application of graph neural networks as emulators for ISSM unstructured-grid simulations of Helheim Glacier, showing that the GNNs are able to accurately emulate the model outputs. The authors then go on to show that the much quicker run time of the emulator allows them to easily determine the required sigma_max parameter in a von Mises calving law to reproduce observed calving-front positions at Helheim between 2007 and 2020.

I congratulate the authors on the substantial work they put in after the second round of reviews to address the concerns raised. I think the current iteration of the paper is much the better for it. On the machine-learning side, the paper is solid and the modifications the authors have made to the manuscript help the reader to better understand what the emulator can achieve. However, I still have some issues regarding the glaciological side of the paper that I don't think the authors dealt with as convincingly, chiefly around the interpretation of the results and the model validation. Improving the validation probably represents enough work to push this into major rather than minor revisions, but I think it has to be done before I can recommend publication.

Page and line numbers refer to those in the clean version of the submitted manuscript.

**Major Comments**

- Applicability of the emulator: the paper very convincingly shows that the emulator functions well on Helheim over the 2007-2020 period, even with different parameter values to those with which it was trained. However, it still does not provide any indication of whether it would function effectively, without retraining, at a different location (or even Helheim under different climatic and topographical conditions). As the authors say, even if the emulator needs retraining for other settings, it will require far fewer simulations than finding the optimal sigma_max by other means, but I think it's important to show whether the emulator could function more or less out of the box, or would need a decent chunk of work to be usable on another use case. If I wanted to use it somewhere else and it turned out I had to compile my own record of calving-front positions and retrain the emulator, that's clearly something I'd really want to know before deciding to use it. At the moment, given the validation is essentially on the setup used to train it, it doesn't persuade me that it has learned something that really exists outside that model domain
- Glaciological interpretation: This is certainly much better than it was before, but I'm still not convinced. Figure 6 shows that predicted sigma_max does mostly follow the expected behaviour (advancing front = higher sigma and vice versa), but it doesn't always (late 2017, for example), and it would really help if the authors can provide some explanation of this. The underlying data are ISSM simulations, so, if everything is physically meaningful, it should be possible for the authors to provide some explanation for these discrepancies that makes sense. Some possibilities are mentioned in the discussion, but I think the paper really needs to offer something more concrete than a few maybes here.

**Minor Comments**

- p.3, l.63-65: Maybe better rephrased as 'GNN emulators take direct advantage of the unstructured meshes of the Ice-sheet and Sea-level System Model (ISSM), allowing flexible spatial resolution and efficient allocation of computational resources'
- p.3, l.67: I think it would be very helpful to evaluate the model on a different site too
- p.3, l.69: 'advances', not 'advancements'
- Section 5.2 and Table 2: Could the authors provide the various timings in hours and minutes as well in brackets after the time in seconds? '948 seconds', for example, is a sufficiently large number of seconds that I have to work out what it means in minutes, which is a bit of a pain.
- p.16, l.357-360: This paragraph is a bit redundant. The entire premise of this paper is that sigma_max has a substantial impact on the terminus position and that its behaviour has some

physical basis. If it didn't, that would be more noteworthy! This might instead be a good place to note the times when sigma_max and the terminus position don't agree and discuss why (or have a sentence saying this will be discussed more in the next section)

- p.16, l.362: 'migration', not 'mitigation'
- p.16, l.364: Better rephrased as 'the calving front retreats too much such that the modeling results no longer match the observations'
- p.18, l.375: 'how difficult it is for calving to happen'
- p.18, l.380: Delete 'this'
- p.18, l.381: But, if the emulator is trained with spatially uniform sigma_max values, would it be able to meaningfully predict a spatially varying field? How much variation/resolution would be needed? Two, three, four flowlines? More? My point is that this is easy to say, but I'm not convinced it would be easy to implement. I might rephrase this less blithely.
- p.18, ;382-386: Might the authors be able to do some of that here in relation to the deviations between sigma_max and the calving front position? The two lines don't always agree in Figure 6 and this would be the perfect place to explore why. Similarly, Figure 8 shows that the optimal-sigma_max run smooths out a lot of the seasonal variation in ice-front position, though it follows the trend well; discussing why this happens would also be interesting

---

## Author Response (AR4)

Thank you for revising your manuscript according to the requests of the 3rd review round. The main comment did relate to the transferability of the emulator to other settings. You have followed this request and applied your emulator to Pine Island Glacier. I truly appreciate this effort. You forward some performance measures that indicate notably reduced accuracy. To better assess these bulk measures, a supplementary figure of plain-view velocity or thickness maps as well as differences (similar to Figs. 4 and 5) would be beneficial to better assess the implications. Please add such a figure to the supplement and refer to it in the text.

Once this figure is included, I will transfer your manuscript to the TC production office for publication.

Dear Editor,

Thank you for your suggestion about the supplementary figures. We added supplements, including two figures of ice velocity and thickness maps (Figs. S1 and S2; similar to Figs. 4 and 5) and a table of ice velocity and thickness RMSEs (Table S1). We also refer to these figures and table in L417.

Again, we sincerely appreciate all your efforts in processing our manuscript.

Best regards,

Younghyun Koo